# Optimal timing of nirmatrelvir/ritonavir treatment after COVID-19 symptom onset or diagnosis: target trial emulation

Carlos K. H. Wong [1,2,3,4,11], Jonathan J. Lau[1,5,6,11], Ivan C. H. Au[2,6], Kristy T. K. Lau[2], Ivan F. N. Hung [7,8], Malik Peiris [6,9], Gabriel M. Leung [1,5,6] & Joseph T. Wu [1,5,6,10]

Reports of symptomatic rebound and/or test re-positivity among COVID-19 patients following the standard five-day treatment course of nirmatrelvir/ritonavir have sparked debates regarding optimal treatment timing and dosage. It is unclear whether initiating nirmatrelvir/ritonavir immediately after symptom onset would improve clinical outcomes and/or lead to post-treatment viral burden rebound due to inadequate viral clearance during treatment. Here we show that, by emulating a randomized target trial using real-world electronic medical record data from all 87,070 adult users of nirmatrelvir/ritonavir in Hong Kong between 16th March 2022 and 15th January 2023, early initiation of nirmatrelvir/ritonavir treatment (0 to 1 days after symptom onset or diagnosis) significantly reduced the incidence of 28-day all-cause mortality and hospitalization compared to delayed initiation (2 or more days) (absolute risk reduction [ARR]: 1.50% (95% confidence interval 1.17-1.80%); relative risk [RR]: 0.77 (0.73, 0.82)), but may be associated with a significant elevated risk of viral burden rebound (ARR: −1.08% (−1.55%, −0.46%)), although the latter estimates were associated with high uncertainty due to limited sample sizes. As such, patients should continue to initiate nirmatrelvir/ritonavir early after symptom onset or diagnosis to better protect against the more serious outcomes of hospitalization and mortality.

Numerous studies have demonstrated the clinical benefit of timely COVID-19 treatment using nirmatrelvir/ritonavir amongst patients with risk factors for progression to severe disease. If used within 5 days of diagnosis or symptom onset, nirmatrelvir/ritonavir significantly reduced the risk of hospitalization or severe disease and death amongst high-risk unvaccinated patients[1] in its licensing EPIC-HR trial, amongst unvaccinated or partially vaccinated patients in hospitalized[2] and ambulatory settings[3], amongst individuals fully vaccinated more

[1]Laboratory of Data Discovery for Health (D24H), Hong Kong SAR, China. [2]Department of Pharmacology and Pharmacy, LKS Faculty of Medicine, The University of Hong Kong, Hong Kong SAR, China. [3]Department of Family Medicine and Primary Care, School of Clinical Medicine, LKS Faculty of Medicine, The University of Hong Kong, Hong Kong SAR, China. [4]Vaccine Confidence Project, Department of Infectious Disease Epidemiology, London School of Hygiene and Tropical Medicine, London, UK. [5]WHO Collaborating Centre for Infectious Disease Epidemiology and Control, School of Public Health, LKS Faculty of Medicine, The University of Hong Kong, Hong Kong SAR, China. [6]School of Public Health, LKS Faculty of Medicine, The University of Hong Kong, Hong Kong SAR, China. [7]Infectious Diseases Division, Department of Medicine, School of Clinical Medicine, LKS Faculty of Medicine, The University of Hong Kong, Hong Kong SAR, China. [8]State Key Laboratory of Emerging Infectious Diseases, Carol Yu Centre for Infection, Department of Microbiology, LKS Faculty of Medicine, The University of Hong Kong, Hong Kong SAR, China. [9]Centre for Immunology and Infection, Hong Kong SAR, China. [10]The University of Hong Kong—Shenzhen Hospital, Shenzhen, China. [11]These authors contributed equally: Carlos K. H. Wong, Jonathan J. Lau. ✉e-mail: joewu@hku.hk

than 20 weeks away[4], amongst individuals reinfected with SARS-CoV-2[5], and against the Omicron BA.2[2,3], BA.4, or BA.5 subvariants[6]. However, very few real-world studies specifically investigated whether initiating nirmatrelvir/ritonavir earlier or later within this 5-day time frame would affect its effectiveness against subsequent hospitalization and/or death.

Further, many case reports and studies have reported symptom recurrence or viral burden rebound (VBR) after initial recovery upon completing a standard 5-day course of nirmatrelvir/ritonavir[7–9]. It has been hypothesized that initiating nirmatrelvir/ritonavir too early after symptom onset or diagnosis may, in some cases, be associated with VBR and symptom recurrence[7,8]. Due to early suppression of viral replication by nirmatrelvir/ritonavir, the host adaptive immune response may not have had sufficient stimulus and time to fully develop by the end of antiviral therapy[10,11], thus allowing any remnant virus to subsequently resume replication[12] and shedding[13]. However, this hypothesis has not hitherto been tested against real-world clinical data.

In this work, we show that early initiation of nirmatrelvir/ritonavir treatment (0–1 days after symptom onset "or diagnosis) significantly reduced the incidence of 28-day all-cause mortality and hospitalization as compared to delayed initiation (2 or more days) (absolute risk reduction [ARR]: 1.50% (95% confidence interval 1.17–1.80%); relative risk [RR]: 0.77 (0.73, 0.82)). However, early initiation may also be associated with significant elevated risk of posttreatment viral burden rebound as compared to delayed initiation (ARR: −1.08% (−1.55%, −0.46%)), although such estimates are based on limited sample sizes and thus subject to high uncertainty.

## Results

We sought to measure the absolute risk reduction [ARR] and relative risk [RR] of 28-day all-cause mortality and hospitalization and VBR attributable to starting nirmatrelvir/ritonavir treatment immediately after symptom onset or diagnosis of COVID-19 versus delayed initiation using real-world evidence. We performed two territory-wide, retrospective cohort analyses using the target trial emulation approach to examine the effect of the timing of nirmatrelvir/ritonavir initiation on the incidence of all-cause mortality, all-cause hospitalization, and VBR among all adult (age ≥18 years) patients who had initiated nirmatrelvir/ritonavir in the Hong Kong Special Administrative Region (SAR), China, between 16th March 2022 (the date when nirmatrelvir/ritonavir were first available for use in Hong Kong) and 15th January 2023. During our study period, COVID-19 infections in Hong Kong were predominately caused by Omicron and its subvariants. The identification of COVID-19 patients eligible for inclusion in the two target trial emulation analyses is illustrated in Fig. 1. These cohorts comprised both outpatients and inpatients who had confirmed diagnoses of COVID-19 and received nirmatrelvir/ritonavir treatment from public clinics or hospitals during the study period. Early initiation was defined as a prescription of nirmatrelvir/ritonavir within 1 day from the date of COVID-19 diagnosis or first symptom onset (day 0–1). Late initiation was defined as prescription on or beyond days 2. Index date was defined as that of SARS-CoV-2 infection diagnosis or symptom onset, whichever occurred earlier.

Briefly, baseline characteristics between the early and late initiation groups were first balanced using inverse probability weightings (IPW) constructed via propensity scores[5]. We then cloned each participant across both initiation groups starting from time zero (i.e., the date of symptoms onset or diagnosis). Clones were then censored on the day when they no longer conformed to their treatment group assignment. To address selection bias from artificial censoring, on each day, uncensored clones were weighted by the inverse of their probability of remaining uncensored[14]. We then calculated nonparametric Kaplan–Meier estimates of the absolute risk reduction (ARR) and relative risk (RR) of mortality and hospitalization, and the ARR of

VBR due to early nirmatrelvir/ritonavir initiation. As very few VBR events were observed amongst those who initiated nirmatrelvir/ritonavir 2 or more days after index date, we were unable to report estimates of the relative risk of VBR due to early initiation. See Methods for details on data and methodology.

First, we showed that early initiation of nirmatrelvir/ritonavir was associated with reduced incidence of all-cause mortality and hospitalization within 28 days of posttreatment follow-up. A total of 87,070 nirmatrelvir/ritonavir users who had confirmed diagnoses of SARS-CoV-2 infection between 16th March 2022 and 15th January 2023 were included in this analysis. Baseline covariates of those with early and late initiation are presented in Table 1. The median duration from COVID-19 diagnosis or first symptom onset to nirmatrelvir/ritonavir initiation was 1 (interquartile range: 0–1) days for early users and 3 (2–3) days for late users. Crude cumulative incidence of all-cause mortality or hospitalization occurring within 28 days of follow-up was 5.09% for early users and 6.59% for late users. Early initiation of nirmatrelvir/ritonavir (i.e., within 1 day of symptom onset or diagnosis) was associated with a significantly lower risk of 28-day all-cause mortality or hospitalization (ARR: 1.50%, 95% CI (1.17%, 1.80%), RR: 0.77, 95% CI (0.73, 0.82)) versus late initiation (2 or more days after symptom onset or diagnosis) (Table 2). A significantly lower risk was also observed if nirmatrelvir/ritonavir was initiated within 2 days of symptom onset or diagnosis versus 3 or more days (ARR: 2.25% (1.40%, 3.02%), RR: 0.70 (0.63, 0.79)), and within 3 days versus 4 or more days (ARR: 2.82% (0.65%, 5.40%), RR: 0.66 (0.50, 0.90)). Cumulative incidence plots for the study outcome of 28-day all-cause mortality or hospitalization among early (0–1 day) and late (2 or more days) initiators are illustrated in Supplementary Fig. 1. Such risk reduction was also consistently observed across patient subgroups by sex, vaccination status, concomitant corticosteroid use, immunocompromised status, or whether the date of symptom onset or diagnosis was used as the index date (Table 2). Results from sensitivity analyses including deaths and hospitalizations up to 42 days post treatment, excluding those initiating nirmatrelvir/ritonavir beyond 5 days after diagnosis or symptom onset were consistent with our main results (Table 2). The exception−a nonsignificant increase in risk between March and June 2022−was likely not representative of the true effect from early initiation due to the small sample size and extraordinary stress faced by the local health system during this time period.

Next, we showed that early initiation of nirmatrelvir/ritonavir (0–1 days after symptom onset or diagnosis) was significantly associated with an elevated risk of VBR compared to late initiation (on or beyond 2 days) (ARR: −1.08% (−1.55%, −0.46%)). Our analysis included patients who had at least one Ct value measurement via real-time quantitative reverse-transcription polymerase chain reaction (RT-qPCR) within 14 days prior to the index date. We also found that early initiation of nirmatrelvir/ritonavir was consistently associated with an elevated risk of VBR for all patient subgroups and across all time periods, with most associations reaching statistical significance. The Ct value trajectory plot of nirmatrelvir/ritonavir users over the first 28 days of follow-up is illustrated in Supplementary Fig. 2 by early or late initiation of the oral antiviral. Consistent with our findings above, the Ct value trajectory plot revealed an increase in the mean viral load (after an initial reduction) amongst early initiators during days 6–8 that was not observed amongst late initiators. Cumulative incidence plots for the study outcome of VBR among early and late initiators are presented in Supplementary Fig. 3.

Meanwhile, initiating nirmatrelvir/ritonavir between 0 and 2 days or 0 and 3 days after the index date were also associated with significant reductions in VBR risk overall versus delaying initiation until 3 or more days or 4 or more days, respectively (Table 3). Overall, the above analysis suggests that early initiation of nirmatrelvir/ritonavir may be associated with an increased risk of VBR, but the validity of such associations is subject to high uncertainty

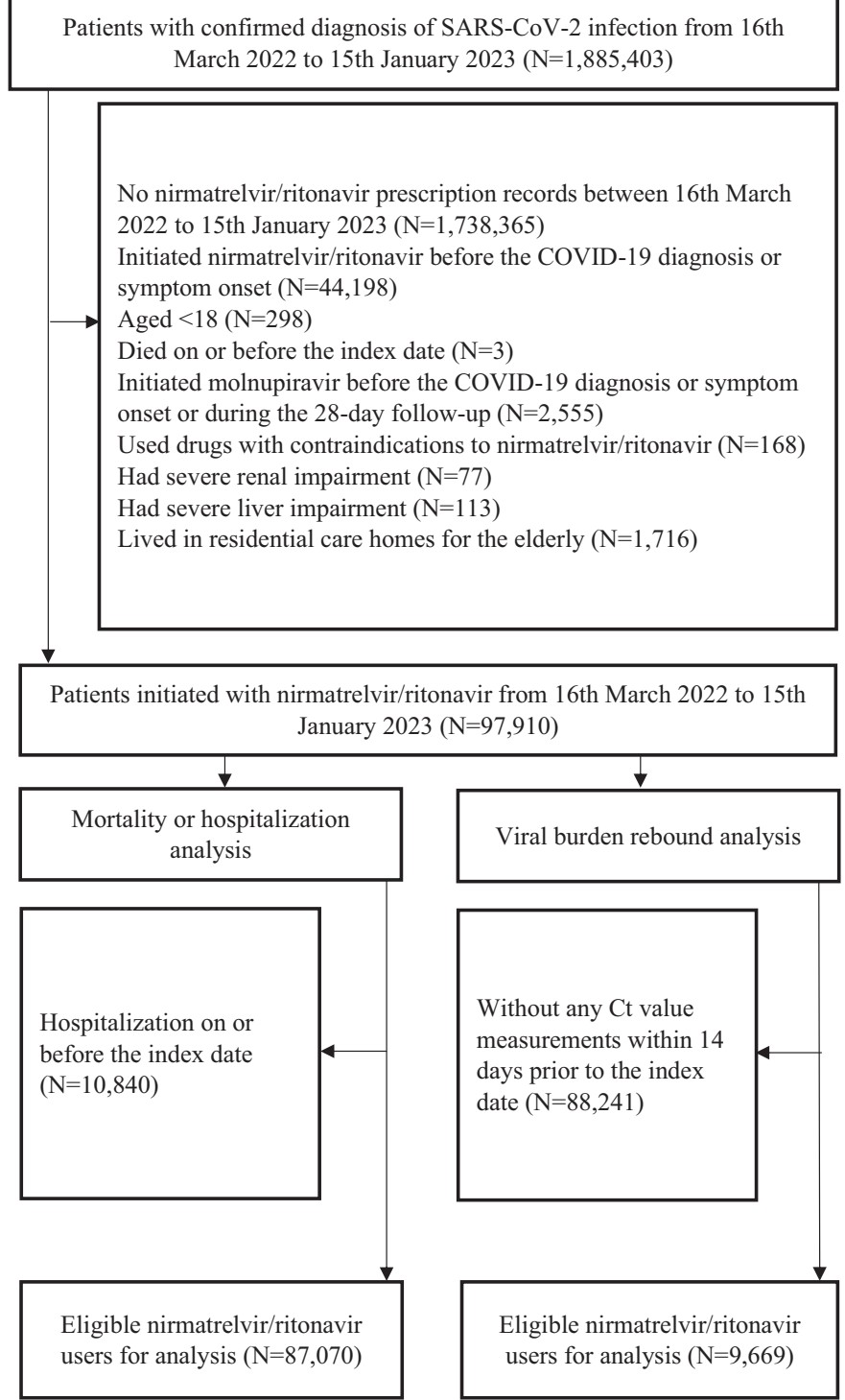

**Fig. 1 | Study flowchart of identifying eligible COVID-19 patients and nirma-trelvir/ritonavir users for inclusion in the two target trial emulations.** Adult nirmatrelvir/ritonavir users with confirmed SARS-CoV-2 infection diagnosis during the study period were identified from the data source. Following the application of study inclusion and exclusion criteria, 87,070 nirmatrelvir/ritonavir users were included for the mortality or hospitalization analysis and 9669 nirmatrelvir/rito-navir users were included for the viral burden rebound analysis.

due to the lack of VBR events amongst those initiating 3 days or greater after the index date.

## Discussion

Our study provides real-world evidence that early initiation of nirma-trelvir/ritonavir is associated with a statistically significant reduction in the risk of 28-day all-cause hospitalization and mortality compared to late initiation. Our findings correspond to earlier indicative findings from ref. 15 that nirmatrelvir/ritonavir was 89.6% effective in pre-venting 30-day hospitalization and death if dispensed on the day of a positive test within 5 days of symptom onset, versus 79.6% if dispensed anytime within 5 days of symptom onset, during a period of BA.4/BA.5 dominance. Our findings also demonstrate that the benefit from early nirmatrelvir/ritonavir treatment is agnostic of sex, age, vaccination

**Table 1 | Baseline characteristics of nirmatrelvir/ritonavir users who initiated early (day 0–1) and late (days ≥ 2) in (a) mortality or hospitalization analysis and (b) viral burden rebound analysis**

| Baseline characteristics | (a) Mortality or hospitalization analysis | | | | | | | | |
| --- | --- | --- | --- | --- | --- | --- | --- | --- | --- |
| | Before IPW | | | | | After IPW | | | |
| | Day 0–1 (N = 43,625) | | Days ≥2 (N = 43,445) | | SMD | Day 0–1 (N = 43,625) | | Days ≥2 (N = 43,445) | | SMD |
| | N/mean | %/SD | N/mean | % / SD | | %/mean | %/SD | %/mean | %/SD | |
| **Age, years** | 62.6 | 14.4 | 61.0 | 15.1 | 0.11 | 61.8 | 14.7 | 61.8 | 14.9 | 0.00 |
| 18–40 | 4232 | 9.7% | 5200 | 12.0% | 0.09 | 10.9% | | 10.7% | | 0.02 |
| 41–65 | 19,267 | 44.2% | 19,812 | 45.6% | | 45.5% | | 44.7% | | |
| >65 | 20,126 | 46.1% | 18,433 | 42.4% | | 43.5% | | 44.6% | | |
| **Sex** | | | | | | | | | | |
| Male | 19,218 | 44.1% | 18,486 | 42.6% | 0.03 | 43.5% | | 43.7% | | 0.00 |
| Female | 24,407 | 55.9% | 24,959 | 57.4% | | 56.5% | | 56.3% | | |
| **Regions** | | | | | | | | | | |
| Hong Kong Island | 7446 | 17.1% | 7369 | 17.0% | 0.01 | 17.0% | | 17.1% | | 0.00 |
| Kowloon | 11,422 | 26.2% | 11,560 | 26.6% | | 26.3% | | 26.2% | | |
| New Territories | 24,718 | 56.7% | 24,469 | 56.3% | | 56.6% | | 56.6% | | |
| Others | 39 | 0.1% | 47 | 0.1% | | 0.1% | | 0.1% | | |
| **Charlson's index** | 2.7 | 1.4 | 2.6 | 1.4 | 0.10 | 2.7 | 1.4 | 2.7 | 1.4 | 0.01 |
| 0–4 | 42,202 | 96.7% | 41,828 | 96.3% | 0.03 | 96.6% | | 95.9% | | 0.03 |
| 5–6 | 1203 | 2.8% | 1,371 | 3.2% | | 2.9% | | 3.4% | | |
| 7–14 | 220 | 0.5% | 246 | 0.6% | | 0.5% | | 0.6% | | |
| Symptomatic presentation | 29,159 | 66.8% | 39,615 | 91.2% | 0.63 | 79.0% | | 78.9% | | 0.00 |
| Concomitant corticosteroid use | 425 | 1.0% | 499 | 1.1% | 0.02 | 1.1% | | 1.2% | | 0.01 |
| Immunocompromised | 603 | 1.4% | 720 | 1.7% | 0.02 | 1.6% | | 1.7% | | 0.00 |
| Healthcare utilization | 1,761 | 4.0% | 1,878 | 4.3% | 0.01 | 4.4% | | 4.7% | | 0.01 |
| Previous SARS-CoV-2 infection | 90 | 0.2% | 29 | 0.1% | 0.04 | 0.1% | | 0.1% | | 0.00 |
| **COVID-19 vaccination status*** | | | | | | | | | | |
| Not fully vaccinated | 5073 | 11.6% | 5661 | 13.0% | 0.05 | 12.5% | | 12.7% | | 0.01 |
| Fully vaccinated but not boosted | 14,883 | 34.1% | 14,966 | 34.4% | | 34.3% | | 34.4% | | |
| Boosted | 23,669 | 54.3% | 22,818 | 52.5% | | 53.2% | | 53.0% | | |
| **Time of SARS-CoV-2 infection diagnosis** | | | | | | | | | | |
| March–June 2022⁺ | 2900 | 6.6% | 2753 | 6.3% | 0.03 | 6.6% | | 6.8% | | 0.01 |
| July–October 2022⁺ | 16,639 | 38.1% | 17,119 | 39.4% | | 38.9% | | 39.0% | | |
| November 2022–January 2023⁺ | 24,086 | 55.2% | 23,573 | 54.3% | | 54.5% | | 54.2% | | |
| **Day of the week of SARS-CoV-2 infection diagnosis** | | | | | | | | | | |
| Sunday | 6084 | 13.9% | 6136 | 14.1% | 0.02 | 14.0% | | 14.3% | | 0.02 |
| Monday | 5812 | 13.3% | 5929 | 13.6% | | 13.4% | | 13.6% | | |
| Tuesday | 5649 | 12.9% | 5608 | 12.9% | | 13.0% | | 12.7% | | |
| Wednesday | 6316 | 14.5% | 6185 | 14.2% | | 14.4% | | 14.4% | | |
| Thursday | 6751 | 15.5% | 6549 | 15.1% | | 15.4% | | 15.0% | | |
| Friday | 6602 | 15.1% | 6441 | 14.8% | | 15.2% | | 14.9% | | |
| Saturday | 6411 | 14.7% | 6597 | 15.2% | | 14.6% | | 15.0% | | |
| **Day of the week of SARS-CoV-2 infection diagnosis** | | | | | | | | | | |
| Weekend | 12,495 | 28.6% | 12,733 | 29.3% | 0.01 | 28.6% | | 29.4% | | 0.02 |
| Weekdays | 31,130 | 71.4% | 30,712 | 70.7% | | 71.4% | | 70.6% | | |
| **Type of viral test for case detection** | | | | | | | | | | |
| Rapid antigen test | 30,957 | 71.0% | 31,833 | 73.3% | 0.05 | 72.2% | | 72.4% | | 0.01 |
| RT-qPCR | 12,668 | 29.0% | 11,612 | 26.7% | | 27.8% | | 27.6% | | |

| Baseline characteristics | (b) Viral burden rebound analysis | | | | | | | | |
| --- | --- | --- | --- | --- | --- | --- | --- | --- | --- |
| | Before IPW | | | | | After IPW | | | |
| | Day 0–1 (N = 9,128) | | Days ≥2 (N = 541) | | SMD | Day 0–1 (N = 9128) | | Days ≥2 (N = 541) | | SMD |
| | N/mean | %/SD | N/mean | %/SD | | %/mean | %/SD | %/mean | %/SD | |
| **Age, years** | 73.0 | 15.2 | 69.4 | 17.0 | 0.24 | 72.8 | 15.3 | 72.5 | 15.6 | 0.02 |
| 18–40 | 412 | 4.5% | 48 | 8.9% | 0.21 | 4.7% | | 5.5% | | 0.04 |
| 41–65 | 1824 | 20.0% | 126 | 23.3% | | 20.2% | | 19.2% | | |
| >65 | 6892 | 75.5% | 367 | 67.8% | | 75.0% | | 75.3% | | |

**Table 1 (continued) | Baseline characteristics of nirmatrelvir/ritonavir users who initiated early (day 0–1) and late (days ≥ 2) in (a) mortality or hospitalization analysis and (b) viral burden rebound analysis**

| Baseline characteristics | (b) Viral burden rebound analysis | | | | | | | | |
| --- | --- | --- | --- | --- | --- | --- | --- | --- | --- |
| | Before IPW | | | | | After IPW | | | |
| | Day 0–1 (N = 9,128) | | Days ≥2 (N = 541) | | SMD | Day 0–1 (N = 9128) | | Days ≥2 (N = 541) | | SMD |
| | N/mean | %/SD | N/mean | %/SD | | %/mean | %/SD | %/mean | %/SD | |
| **Sex** | | | | | | | | | | |
| Male | 4837 | 53.0% | 296 | 54.7% | 0.03 | 53.1% | | 53.8% | | 0.01 |
| Female | 4291 | 47.0% | 245 | 45.3% | | 46.9% | | 46.2% | | |
| **Regions** | | | | | | | | | | |
| Hong Kong Island | 1867 | 20.5% | 104 | 19.2% | 0.07 | 20.4% | | 19.8% | | 0.02 |
| Kowloon | 2526 | 27.7% | 145 | 26.8% | | 27.6% | | 28.1% | | |
| New Territories | 4693 | 51.4% | 287 | 53.0% | | 51.5% | | 51.6% | | |
| Others | 42 | 0.5% | 5 | 0.9% | | 0.5% | | 0.5% | | |
| **Charlson's index** | 4.6 | 1.8 | 4.3 | 2.1 | 0.16 | 4.5 | 1.8 | 4.5 | 1.9 | 0.01 |
| 0–4 | 4270 | 46.8% | 285 | 52.7% | 0.13 | 47.1% | | 48.0% | | 0.04 |
| 5–6 | 3808 | 41.7% | 193 | 35.7% | | 41.5% | | 39.5% | | |
| 7–14 | 1050 | 11.5% | 63 | 11.6% | | 11.4% | | 12.4% | | |
| Symptomatic presentation | 968 | 10.6% | 74 | 13.7% | 0.09 | 10.8% | | 10.5% | | 0.01 |
| Inpatient users | 8861 | 97.1% | 519 | 95.9% | 0.06 | 97.0% | | 97.1% | | 0.01 |
| Concomitant corticosteroid use | 1976 | 21.6% | 128 | 23.7% | 0.05 | 21.8% | | 22.9% | | 0.02 |
| Immunocompromised | 2112 | 23.1% | 108 | 20.0% | 0.08 | 23.0% | | 23.6% | | 0.01 |
| Healthcare utilization | 6045 | 66.2% | 342 | 63.2% | 0.06 | 66.1% | | 65.5% | | 0.01 |
| Previous SARS-CoV-2 infection | 1294 | 14.2% | 77 | 14.2% | 0.00 | 14.2% | | 14.8% | | 0.02 |
| **COVID-19 vaccination status*** | | | | | | | | | | |
| Not fully vaccinated | 2,762 | 30.3% | 197 | 36.4% | 0.13 | 30.6% | | 31.5% | | 0.02 |
| Fully vaccinated but not boosted | 4,028 | 44.1% | 216 | 39.9% | | 43.9% | | 42.7% | | |
| Boosted | 2,338 | 25.6% | 128 | 23.7% | | 25.5% | | 25.8% | | |
| **Time of SARS-CoV-2 infection diagnosis** | | | | | | | | | | |
| March–June 2022[†] | 697 | 7.6% | 100 | 18.5% | 0.33 | 8.3% | | 8.7% | | 0.03 |
| July–October 2022[†] | 4495 | 49.2% | 250 | 46.2% | | 49.1% | | 47.7% | | |
| November 2022–January 2023[†] | 3936 | 43.1% | 191 | 35.3% | | 42.7% | | 43.6% | | |
| **Day of the week of SARS-CoV-2 infection diagnosis** | | | | | | | | | | |
| Sunday | 1314 | 14.4% | 68 | 12.6% | 0.19 | 14.4% | | 13.5% | | 0.20 |
| Monday | 1322 | 14.5% | 67 | 12.4% | | 14.5% | | 11.6% | | |
| Tuesday | 1260 | 13.8% | 68 | 12.6% | | 13.8% | | 12.4% | | |
| Wednesday | 1273 | 13.9% | 70 | 12.9% | | 14.0% | | 13.1% | | |
| Thursday | 1350 | 14.8% | 70 | 12.9% | | 14.8% | | 12.5% | | |
| Friday | 1349 | 14.8% | 89 | 16.5% | | 14.8% | | 16.9% | | |
| Saturday | 1260 | 13.8% | 109 | 20.1% | | 13.8% | | 20.0% | | |
| **Day of the week of SARS-CoV-2 infection diagnosis** | | | | | | | | | | |
| Weekend | 2574 | 28.2% | 177 | 32.7% | 0.10 | 28.2% | | 33.5% | | 0.12 |
| Weekdays | 6554 | 71.8% | 364 | 67.3% | | 71.8% | | 66.5% | | |
| **Type of viral test for case detection** | | | | | | | | | | |
| Rapid antigen test | 725 | 7.9% | 26 | 4.8% | 0.13 | 7.8% | | 8.6% | | 0.03 |
| RT-qPCR | 8403 | 92.1% | 515 | 95.2% | | 92.2% | | 91.4% | | |
| Presence of at least one Ct value measurement within five days from diagnosis or symptom onset | 8483 | 92.9% | 506 | 93.5% | 0.02 | 93.0% | | 93.3% | | 0.01 |

IPW inverse probability weighting, SD standard deviation, SMD standardized mean difference, NA not applicable, RT-qPCR real-time quantitative reverse-transcription polymerase chain reaction.
*Fully vaccinated but not boosted patients were defined as those with two doses of BNT162b2 (Comirnaty) or three doses of COVID-19 Vaccine (Vero Cell), Inactivated (CoronaVac); boosted patients were defined as those with at least three doses of BNT162b2 (Comirnaty) or four doses of COVID-19 Vaccine (Vero Cell), Inactivated (CoronaVac).
[†]Omicron subvariant BA.2 was dominant from March to June 2022, BA.2 and BA.4/BA.5 from July to October 2022 and BA.5, BA.2.75, and BQ.1 from November 2022 to January 2023.

status, concomitant corticosteroid use, primary immunodeficiencies or the then dominant Omicron subvariant (BA.4/BA.5 between July and October 2022, and a mix of BA.5, BA.2.75 and BQ.1 subvariants between November 2022 and January 2023).

The observed benefit from administering nirmatrelvir/ritonavir as soon as possible after diagnosis or symptom onset may be attributable to an early suppression of viral replication, which in turn reduced the risk of subsequent deterioration to severe disease and death. Early viral

**Table 2 | Association between timing of nirmatrelvir/ritonavir initiation and 28-day all-cause mortality or hospitalization**

| All-cause mortality or hospitalization | Cumulative incidence | | | | Absolute risk reduction[a] | 95% CI | Relative risk[a] | 95% CI |
|---|---|---|---|---|---|---|---|---|
| | Early | | Late | | | | | |
| | N | Risk[Δ] | N | Risk[Δ] | | | | |
| **Primary analysis** | | | | | | | | |
| Target trial emulation with IPW | 43,625 | 5.09% | 43,445 | 6.59% | 1.50% | (1.17%, 1.80%) | 0.77 | (0.73, 0.82) |
| **Sensitivity analyses** | | | | | | | | |
| Extended the follow-up to 42 days | 43,625 | 5.20% | 43,445 | 6.70% | 1.50% | (1.14%, 1.81%) | 0.78 | (0.74, 0.82) |
| Days 0–2 vs days >2 | 64,690 | 5.24% | 22,380 | 7.50% | 2.25% | (1.40%, 3.02%) | 0.70 | (0.63, 0.79) |
| Days 0–3 vs days >3 | 76,935 | 5.54% | 10,135 | 8.36% | 2.82% | (0.65%, 5.40%) | 0.66 | (0.50, 0.90) |
| Exclude those who initiated nirmatrelvir/ritonavir beyond 5 days after diagnosis or symptom onset | 43,625 | 4.71% | 41,686 | 6.81% | 2.10% | (1.77%, 2.43%) | 0.69 | (0.66, 0.73) |
| **Subgroup analyses** | | | | | | | | |
| Male | 19,218 | 5.75% | 18,486 | 7.55% | 1.80% | (1.32%, 2.21%) | 0.76 | (0.72, 0.82) |
| Female | 24,407 | 4.61% | 24,959 | 5.82% | 1.21% | (0.78%, 1.55%) | 0.79 | (0.74, 0.86) |
| March–June 2022[†] | 2900 | 13.13% | 2,753 | 11.63% | −1.50% | (−2.83%, 0.48%) | 1.13 | (0.96, 1.25) |
| July–October 2022[†] | 16,639 | 6.19% | 17,119 | 8.45% | 2.26% | (1.78%, 2.73%) | 0.73 | (0.69, 0.78) |
| November 2022–January 2023[†] | 24,086 | 3.52% | 23,573 | 4.89% | 1.37% | (0.82%, 1.82%) | 0.72 | (0.65, 0.81) |
| Fully vaccinated or boosted | 38,552 | 4.02% | 37,784 | 5.55% | 1.53% | (1.23%, 1.91%) | 0.72 | (0.67, 0.77) |
| Not fully vaccinated | 5073 | 12.20% | 5661 | 13.67% | 1.47% | (0.53%, 2.39%) | 0.89 | (0.83, 0.96) |
| Charlson's index 0–6 | 43,405 | 4.62% | 43,199 | 6.04% | 1.42% | (1.08%, 1.76%) | 0.77 | (0.72, 0.82) |
| Charlson's index >6 | 220 | 97.51% | 246 | 97.97% | 0.46% | (−2.93%, 2.81%) | 1.00 | (0.97, 1.03) |
| Concomitant corticosteroid use | 425 | 95.51% | 499 | 98.48% | 2.97% | (1.07%, 4.92%) | 0.97 | (0.95, 0.99) |
| No concomitant corticosteroid use | 43,200 | 4.11% | 42,946 | 5.50% | 1.38% | (1.09%, 1.65%) | 0.75 | (0.71, 0.79) |
| Immunocompromised | 603 | 74.74% | 720 | 83.84% | 9.10% | (4.55%, 14.89%) | 0.89 | (0.82, 0.95) |
| Not immunocompromised | 43,022 | 3.98% | 42,725 | 5.20% | 1.22% | (0.91%, 1.57%) | 0.77 | (0.71, 0.82) |
| With at least one Ct value measurement | 2398 | 70.20% | 2871 | 85.36% | 15.16% | (13.07%, 17.22%) | 0.82 | (0.80, 0.85) |
| Without any Ct value measurements | 41,227 | 0.64% | 40,574 | 0.72% | 0.09% | (−0.08%, 0.25%) | 0.88 | (0.72, 1.14) |
| Documented symptom onset date as index date | 29,159 | 4.70% | 39,615 | 6.11% | 1.42% | (1.12%, 1.71%) | 0.77 | (0.72, 0.81) |
| Date of COVID-19 diagnosis as index date | 14,534 | 4.15% | 3870 | 5.45% | 1.31% | (0.76%, 1.87%) | 0.76 | (0.69, 0.85) |

IPW inverse probability weighting, CI confidence interval.

[Δ]Risk represents the incidence of 28-day all-cause mortality or hospitalization after inverse probability weighting (IPW) between the two treatment groups against baseline covariates.

[a]Absolute risk reduction >0 (or <0) and relative risk <1 (or >1) indicate early initiators (day 0–1) had a lower (higher) risk of the designated outcome compared to late initiators (days ≥2).

[†]Omicron subvariant BA.2 was dominant from March to June 2022, BA.2 and BA.4/BA.5 from July to October 2022 and BA.5, BA.2.75, and BQ.1 from November 2022 to January 2023.

**Table 3 | Association between timing of nirmatrelvir/ritonavir initiation and 28-day viral burden rebound**

| All-cause mortality or hospitalization | Cumulative incidence | | | | Absolute risk reduction[a] | 95% CI |
|---|---|---|---|---|---|---|
| | Early | | Late | | | |
| | N | Risk[Δ] | N | Risk[Δ] | | |
| **Primary analysis (days 0–1 vs days > 1)** | | | | | | |
| Overall | 9128 | 1.39% | 541 | 0.31% | −1.08% | (−1.55%, −0.46%) |
| *Subgroup analyses* | | | | | | |
| Male | 4837 | 1.62% | 296 | 0.64% | −0.99% | (−1.81%, 0.32%) |
| Female | 4291 | 1.12% | 245 | 0.00% | −1.12% | (−1.41%, −0.88%) |
| March–June 2022[†] | 697 | 0.59% | 100 | 0.00% | −0.59% | (−1.20%, −0.15%) |
| July–October 2022[†] | 4495 | 1.88% | 250 | 0.57% | −1.31% | (−2.18%, −0.06%) |
| November 2022–January 2023[†] | 3936 | 0.98% | 191 | 0.18% | −0.80% | (−1.18%, −0.36%) |
| Fully vaccinated or boosted | 6366 | 1.06% | 344 | 0.43% | −0.62% | (−1.10%, 0.25%) |
| Not fully vaccinated | 2762 | 2.14% | 197 | 0.00% | −2.14% | (−2.52%, −1.69%) |
| Charlson's index 0–6 | 8078 | 1.24% | 478 | 0.36% | −0.88% | (−1.37%, −0.27%) |
| Charlson's index >6 | 1047 | 2.57% | 63 | 0.00% | −2.57% | (−3.48%, −1.68%) |
| Concomitant corticosteroid use | 1976 | 2.20% | 128 | 0.00% | −2.20% | (−2.83%, −1.55%) |
| No concomitant corticosteroid use | 7152 | 1.17% | 413 | 0.40% | −0.77% | (−1.35%, −0.12%) |
| Immunocompromised | 2112 | 1.86% | 108 | 0.00% | −1.86% | (−2.57%, −1.42%) |
| Not immunocompromised | 7016 | 1.25% | 433 | 0.43% | −0.83% | (−1.39%, −0.03%) |
| Documented symptom onset date as index date | 968 | 1.42% | 74 | 1.36% | −0.06% | (−1.95%, 4.67%) |
| Date of COVID-19 diagnosis as index date | 8571 | 1.38% | 501 | 0.14% | −1.24% | (−1.58%, −0.83%) |
| Inpatient users | 8861 | 1.40% | 519 | 0.29% | −1.11% | (−1.56%, −0.55%) |
| Outpatient users | 267 | 1.18% | 22 | 0.00% | −1.18% | (−2.78%, 0.00%) |
| **Sensitivity analysis (days 0–2 vs days > 2)** | | | | | | |
| Overall | 9452 | 1.38% | 217 | 0.00% | −1.38% | (−1.67%, −1.18%) |
| *Subgroup analyses* | | | | | | |
| Male | 5011 | 1.62% | 122 | 0.00% | −1.62% | (−2.15%, −1.26%) |
| Female | 4441 | 1.11% | 95 | 0.00% | −1.11% | (−1.41%, −0.86%) |
| March–June 2022[†] | 756 | 0.57% | 41 | 0.00% | −0.57% | (−1.32%, 0.00%) |
| July–October 2022[†] | 4645 | 1.86% | 100 | 0.00% | −1.86% | (−2.20%, −1.47%) |
| November 2022–January 2023[†] | 4051 | 0.98% | 76 | 0.00% | −0.98% | (−1.30%, −0.66%) |
| Fully vaccinated or boosted | 6567 | 1.06% | 143 | 0.00% | −1.06% | (−1.30%, −0.81%) |
| Not fully vaccinated | 2885 | 2.11% | 74 | 0.00% | −2.11% | (−2.54%, 0.00%) |
| Charlson's index 0–6 | 8363 | 1.24% | 193 | 0.00% | −1.24% | (−1.47%, −1.02%) |
| Charlson's index >6 | 1086 | 2.49% | 24 | 0.00% | −2.49% | (−3.34%, 0.00%) |
| Concomitant corticosteroid use | 2044 | 2.12% | 60 | 0.00% | −2.12% | (−2.72%, −1.45%) |
| No concomitant corticosteroid use | 7408 | 1.18% | 157 | 0.00% | −1.18% | (−1.50%, −0.92%) |
| Immunocompromised | 2173 | 1.79% | 47 | 0.00% | −1.79% | (−2.35%, 0.00%) |
| Not immunocompromised | 7279 | 1.26% | 170 | 0.00% | −1.26% | (−1.50%, −1.03%) |
| Documented symptom onset date as index date | 1018 | 1.50% | 24 | 0.00% | −1.50% | (−2.23%, −0.66%) |
| Date of COVID-19 diagnosis as index date | 8864 | 1.37% | 208 | 0.00% | −1.37% | (−1.63%, −1.11%) |
| Inpatient users | 9173 | 1.39% | 207 | 0.00% | −1.39% | (−1.76%, −1.14%) |
| Outpatient users | 279 | 1.10% | 10 | 0.00% | −1.10% | (−2.91%, 0.00%) |
| **Sensitivity analysis (days 0–3 vs days > 3)** | | | | | | |
| Overall | 9580 | 1.39% | 89 | 0.00% | −1.39% | (−1.68%, −1.19%) |
| *Subgroup analyses* | | | | | | |
| Male | 5081 | 1.63% | 52 | 0.00% | −1.63% | (−2.16%, −1.26%) |
| Female | 4499 | 1.12% | 37 | 0.00% | −1.12% | (−1.41%, −0.86%) |
| March–June 2022[†] | 779 | 0.57% | 18 | 0.00% | −0.57% | (_1.33%, 0.00%) |
| July–October 2022[†] | 4705 | 2.04% | 40 | 0.00% | −2.04% | (−2.45%, −1.63%) |
| November 2022–January 2023[†] | 4096 | 0.97% | 31 | 0.00% | −0.97% | (−1.31%, −0.66%) |
| Fully vaccinated or boosted | 6651 | 1.06% | 59 | 0.00% | −1.06% | (−1.30%, −0.81%) |
| Not fully vaccinated | 2929 | 2.14% | 30 | 0.00% | −2.14% | (−2.58%, 0.00%) |
| Charlson's index 0–6 | 8482 | 1.24% | 74 | 0.00% | −1.24% | (−1.47%, −1.03%) |
| Charlson's index >6 | 1095 | 2.76% | 15 | 0.00% | −2.76% | (−3.81%, 0.00%) |

**Table 3 (continued) | Association between timing of nirmatrelvir/ritonavir initiation and 28-day viral burden rebound**

| All-cause mortality or hospitalization | Cumulative incidence | | | | Absolute risk reduction[a] | 95% CI |
|---|---|---|---|---|---|---|
| | Early | | Late | | | |
| | N | Risk[Δ] | N | Risk[Δ] | | |
| Concomitant corticosteroid use | 2075 | 2.10% | 29 | 0.00% | −2.10% | (−2.78%, −1.37%) |
| No concomitant corticosteroid use | 7505 | 1.18% | 60 | 0.00% | −1.18% | (−1.50%, −0.92%) |
| Immunocompromised | 2200 | 1.79% | 20 | 0.00% | −1.79% | (−2.35%, 0.00%) |
| Not immunocompromised | 7380 | 1.26% | 69 | 0.00% | −1.26% | (−1.50%, −1.03%) |
| Documented symptom onset date as index date | 1034 | 1.48% | 8 | 0.00% | −1.48% | (−2.25%, −0.62%) |
| Date of COVID-19 diagnosis as index date | 8986 | 1.38% | 86 | 0.00% | −1.38% | (−1.64%, −1.11%) |
| Inpatient users | 9294 | 1.40% | 86 | 0.00% | −1.40% | (−1.76%, −1.14%) |
| Outpatient users | 286 | 1.07% | 3 | 0.00% | −1.07% | (−2.85%, 0.00%) |

CI confidence interval.

[Δ]Risk represents the incidence of 28-day viral burden rebound after inverse probability weighting (IPW) between the two treatment groups against baseline covariates.

[a]Absolute risk reduction >0 (or <0) and relative risk <1 (or >1) indicate early initiators (day 0–1) had a lower (higher) risk of the designated outcome compared to late initiators (days ≥2).

[†]Omicron subvariant BA.2 was dominant from March to June 2022, BA.2 and BA.4/BA.5 from July to October 2022 and BA.5, BA.2.75, and BQ.1 from November 2022 to January 2023.

dynamics models predicted that initiating effective antiviral treatment immediately following symptom onset could reduce the duration of detectable viral shedding and limit cytolytic immune responses required to clear infection, both of which may be linked to reduced disease severity[16]. Lower viral loads in the early phase of infection are also associated with a lower risk of severe clinical outcomes[17,18]. In turn, both the EPIC-HR[1] and the EPIC-SR trials[19] demonstrated nirmatrelvir/ritonavir could swiftly reduce viral load after treatment initiation. Thus, the reduced risk of eventual hospitalization and/or death as observed in our study could potentially be explained by the effective inhibition of nascent viral replication via early initiation of nirmatrelvir/ritonavir treatment.

We presented evidence of a statistically significant elevated risk of VBR overall from early initiation of nirmatrelvir/ritonavir as compared to later initiation (albeit with wide confidence intervals and hence high uncertainty due to the paucity of VBR events amongst late initiators). Such findings correspond to results from a small perspective cohort study, which found a statistically insignificant association between nirmatrelvir/ritonavir initiation within 0–1 days after diagnosis and VBR, and no such association for later initiation, amongst 72 ambulatory patients with a median of 4 vaccination doses[20]. The elevated VBR risk observed amongst early initiators overall may have reflected the effects of aging on nirmatrelvir/ritonavir users, many of whom are aged 60 years or above, and whom may be susceptible to immunosenescence, weakened development of T-cell response, and poorer antiviral antibody quality[21,22], in turn hampering the rapid development of an effective immune response to sufficiently address any leftover virus subsequent to therapy cessation.

We previously observed that being fully vaccinated or boosted was associated with higher risks of VBR regardless of the timing of nirmatrelvir/ritonavir initiation[7]. However, in our subgroup analysis (Table 3), the elevated risk of VBR did not reach statistical significance for fully vaccinated or boosted patients who initiated nirmatrelvir/ritonavir within 1 day of the index date. This is consistent with the ability of vaccines to prime for a faster adaptive immune response after infection and accelerate viral clearance[23], reducing the risk that early use of nirmatrelvir/ritonavir could prematurely[7,24] suppress viral load, hinder the development of a robust immune response to the breakthrough pathogen, and result in subsequent VBR.

Meanwhile, the elevated VBR risk amongst immunocompromised early initiators or those concomitantly using corticosteroids is also consistent with the higher risk of VBR[7], poorer adaptive immune response[25], increased risk of severe disease[26] and prolonged viral shedding[27] observed amongst such individuals. The cessation of nirmatrelvir/ritonavir earlier in the disease course due to early initiation

of therapy may thus exacerbate their inability to sufficiently clear any remnant virus, thereby triggering subsequent VBR.

Conversely, some previous studies[11,28] did not identify evidence of delayed or impaired development of humoral and/or T-cell immunity amongst fully vaccinated and boosted patients subsequent to nirmatrelvir/ritonavir administration. However, in the foregoing studies, immunological samples were only collected from two subjects in total prior to or during the start of symptomatic rebound. All other samples were collected at least 2 days after the start date of symptomatic rebound (where documented) or 10+ days after initial diagnosis. Thus, we submit that, while the foregoing studies demonstrated that nirmatrelvir/ritonavir did not impair the eventual development of sterilizing humoral and/or T-cell immunity, they may not have been well-timed to capture the state of the immune response immediately after cessation of nirmatrelvir/ritonavir therapy, or whether such a state may be causally associated with subsequent VBR. Larger studies with sampling during and after therapy but prior to symptomatic rebound (if any) may be necessary to elucidate the pharmacological and immunological pathways responsible for VBR.

Our study has several strengths. We implemented a target trial emulation design, which allowed us to emulate the random assignment of patients[14,28,29] into early and late initiation, and different combinations thereof, of nirmatrelvir/ritonavir treatment, as would be done in randomized trials, and to measure the effect of initiation timing on hospitalization, mortality and VBR. Our study made use of official healthcare databases encompassing all reported cases of confirmed SARS-CoV-2 infection in Hong Kong. Our study period covered multiple waves of infection by a wide variety of Omicron subvariants (BA.2, BA.4/BA.5, BA.2.75, and BQ.1), during which the city experienced varying levels of health system stress and quality of care. Randomized trials of the aforementioned size and duration involving a deliberate delay of antiviral initiation would have been difficult to carry out due to operational or ethical reasons. To mitigate confounding effects, we first balanced both treatment groups in our main and all subgroup analyses using IPW corresponding to key baseline characteristics. Detailed patient and antiviral prescription records permitted adjustments to be made for a wide range of covariates and potential confounders affecting the risk of severe outcomes subsequent to nirmatrelvir/ritonavir treatment. We then followed with a cloning-censoring-weighting approach[30–32] to eliminate immortal time bias stemming from the delay between time zero (date of symptom onset or diagnosis, i.e., when patients become eligible to receive nirmatrelvir/ritonavir) and treatment assignment (when patients actually receive nirmatrelvir/ritonavir)[14]. Further, our analysis required precise classification of patients between the early and late treatment groups. This was made possible by the availability of detailed symptom onset

data for the majority of patients in our study. Further, rapid and high-quality COVID-19 testing via rapid antigen testing (RAT) or RT-qPCR was widely available in all districts of Hong Kong throughout the study period except for the first few weeks during which the first Omicron wave peaked with the healthcare system overwhelmed (see below for further discussion). As a result, the date of diagnosis for most individuals should closely match their date of symptom onset for most cases, even in cases where the data on the latter was not available. This thereby reduced the risk of misclassification bias between the early and late treatment groups due to testing delays. Lastly, to eliminate any use of post-day-zero information in informing the eligibility and selection of participants in the target trial for our VBR analysis, we opted to over-expansively include as subjects all patients with available Ct value measurements within 14 days prior to the index date. We did not restrict our analysis to only those with consecutive Ct value measurements after the index date and for whom the detection of VBR was thus possible, as this would have introduced post-day-zero information into subject selection for our target trial. Further, following our previous work on VBR[7], patients with repeated Ct value measurements (from RT-qPCR testing) were more likely to have been at higher risk of progression to severe disease than those without. Expanding the eligibility criteria of our VBR analysis to all patients with one or more Ct value measurements within 14 days before the index date would thus minimize potential selection bias from selecting only those with repeated Ct value measurements.

However, our study was also subject to several limitations. First, similar to our earlier work[2,3], residents of residential care homes for the elderly, who suffered disproportionately from COVID-19 during the early half of 2022, were excluded from our analysis due to substantial missing records and often complex patient histories. Second, potential delays in the confirmation of infected cases by RT-qPCR tests and patients attending medical visits for oral antiviral prescription could not be ruled out. This might especially be the case at the beginning of our study period (March 2022), when the public healthcare system in Hong Kong was overwhelmed with newly diagnosed COVID-19 cases during the wave peak predominated by Omicron BA.2 infection. Such potential increase in the lag time between case detection and oral antiviral prescription (confounded by accessibility to healthcare services) may also explain the heterogeneity of treatment effects during the period from March to June 2022, which comprised of less 5% of our subjects in both the mortality/hospitalization and VBR analyses. Nonetheless, results from subgroup analyses separately considering subjects with symptom onset dates or dates of COVID-19 diagnosis as index dates (Tables 2 and 3) were consistent with our primary analyses in both emulated target trials. Thus, we surmised that delays in diagnosis only had a limited impact on our analysis. Third, as Ct value measurements were mainly collected from hospitalized patients, and testing frequency might have depended on patient severity, physician clinical judgment, and the availability of healthcare resources; ascertainment bias in the VBR study outcome could not be ruled out[7]. Fourth, information on any recurrence or progression of symptoms after the index date were not available in our data sources; hence, our study focused on the outcome of VBR instead of COVID-19 symptomatic rebound that could have been indicated by the recurrence of symptoms and/or re-positive viral tests. Further, while we acknowledge that clinical progression of symptoms might have had an impact on patient healthcare-seeking behavior and hence the timing of antiviral initiation, we submit that our large set of baseline covariate adjustments adequately balanced the risks of clinical deterioration between the two treatment groups, ensuring exchangeability. Also, given that nirmatrelvir/ritonavir-eligible patients were already of high risk, those experiencing clinical deterioration and thus presenting themselves to public healthcare facilities would have most likely been hospitalized immediately or soon after and be prescribed nirmatrelvir/ritonavir in an inpatient setting. Such patients would thus have been entirely excluded from our hospitalization/mortality study given our exclusion of patients who were hospitalized/died on the index date. Thus, we surmised that the risk of bias due to our lack of clinical progression data was likely immaterial.

Fifth, we were unable to confirm whether outpatient individuals receiving nirmatrelvir/ritonavir adhered to treatment as recommended. Finally, the small number of VBR occurrences in either the early or late initiation groups limited the precision of our estimates, leading to very wide confidence intervals which may have obscured the true effects from early nirmatrelvir/ritonavir initiation, particularly within our subgroup analyses.

In conclusion, our results suggest that initiating nirmatrelvir/ritonavir as early as possible after symptom onset or diagnosis better protects against all-cause mortality and hospitalization than initiating therapy with longer delays. However, earlier initiation may be associated with an elevated risk of VBR. Nonetheless, to best realize the clinical benefits of nirmatrelvir/ritonavir therapy – the prevention of hospitalization, clinical deterioration, and death – high-risk individuals should continue to have ready access to sensitive and specific COVID-19 testing, so that diagnosis could be made as soon as possible after symptom onset. In particular, such individuals should be advised to undergo RAT as soon as possible after developing upper respiratory tract symptoms to confirm COVID-19 infection and to receive early antiviral treatment. Meanwhile, health systems should also continue to promote patient and clinician awareness of the medication and facilitate widespread and equitable[33] access through community test-and-treat programs, so that nirmatrelvir/ritonavir treatment could be initiated as soon as possible following a positive test to improve patient outcomes and reduce the risk of subsequent deterioration into severe disease.

## Methods
### Data sources
Patients with SARS-CoV-2 infection diagnosis were identified from the Centre for Health Protection, Department of Health of the Hong Kong SAR, China, confirmed by a positive RT-qPCR or RAT result. Throughout the study period, residents had free access to RT-qPCR tests via community testing centers (available across all residential districts in Hong Kong); or alternatively, they could self-arrange RAT (test kits were freely distributed by the Government) and report positive results to the Department of Health. Individuals with COVID-19-related symptoms or recent contact with infected patients were encouraged to be tested; and for high-risk groups such as healthcare workers and staff at residential care facilities, regular mandatory testing was required. Up to late 2022, compulsory testing notices were also issued to residents of residential premises visited by tested positive cases or where sewage samples indicated the presence of SARS-CoV-2 viral RNA. Prescription of oral antivirals and disease management of COVID-19 patients were available from all public healthcare facilities (hospitals and outpatient clinics) managed by the Hospital Authority (HA), the statutory body managing all public healthcare services in Hong Kong. Public healthcare facilities are located in all residential districts in Hong Kong and many were open 7 days a week and during all public holidays during our study period.

Electronic medical records of COVID-19 patients (which included demographics, disease diagnoses, drug prescription and dispensing records, laboratory test results, hospital admissions, attendance to accident & emergency department, and inpatient procedures) were retrieved from the HA. The HA data were anonymously linked to individual vaccination records provided by the Department of Health using unique identification numbers. Mortality events both inside and outside the public hospitals were extracted from the Hong Kong Death Registry. These databases have been used in previous studies to evaluate the treatment effects of novel oral antivirals against COVID-19 at the population level[2,3,34].

## Study specification

In this study, we emulated two target trials using observational data[14,28,35] to explore the association between early versus late initiation of nirmatrelvir/ritonavir among COVID-19 patients on, respectively, (i) all-cause mortality or hospitalization and (ii) VBR. Specifications of the main components of the hypothetical target trials and our emulated trials using observational data are detailed in Supplementary Table 1 (on all-cause mortality or hospitalization) and Supplementary Table 2 (on VBR).

Early initiation was defined as prescription of nirmatrelvir/ritonavir within 1 day from the date of COVID-19 diagnosis (first positive RAT or RT-qPCR), or first symptom onset (days 0–1); while late initiation was defined as the prescription of nirmatrelvir/ritonavir 2 or more days after the index date. Index date was set at the date of SARS-CoV-2 infection diagnosis or symptom onset, whichever occurred earlier. We also performed sensitivity analyses classifying early initiation as prescription within 2 days and 3 days from the index date, and late initiation as prescription after days 3 and 4, respectively. Patients were observed from the index date until registered death, outcome event occurrence, 28 days after the index date, or the administrative end of the follow-up period (February 12, 2023), whichever the earliest. For VBR analysis, patients would be censored at registered death (competing event) but not upon hospital admission, and at day 5 when the nirmatrelvir/ritonavir was not prescribed within 5 days from the index date; and Ct value measurements between time zero (index date) and day 28, including those obtained during hospitalization, were included for the detection of VBR.

## Study population

Our study population included all adult patients with confirmed SARS-CoV-2 infection diagnosis between March 16, 2022 (when nirmatrelvir/ritonavir was locally available for prescription) and January 15, 2023 in Hong Kong SAR, China. Infections in the study period were predominated by Omicron and its subvariants. According to the HA clinical management guideline for adult patients with COVID-19[36], patients (1) with mild symptoms, (2) at risk of progressing to severe disease (i.e., diabetes, obesity with a body mass index ≥30 kg/m², age ≥60 years, immunocompromised state, underlying chronic illnesses, or not fully vaccinated), and (3) at an early stage of disease (within five days of symptom onset), were recommended to receive molnupiravir or nirmatrelvir/ritonavir. Later versions (since March 21, 2022) of the guidelines added that nirmatrelvir/ritonavir should be preferentially administered over molnupiravir, unless the patient was on any concomitant medication(s) contraindicated for nirmatrelvir/ritonavir. Nirmatrelvir/ritonavir users were defined as patients who had received 300 mg nirmatrelvir and 100 mg ritonavir twice daily for 5 days, or 150 mg nirmatrelvir and 100 mg ritonavir twice daily for 5 days among those with moderate renal impairment (estimated glomerular filtration rate [eGFR] ≥30 to <60 mL/min/1.73 m²). Nirmatrelvir/ritonavir treatment was administered orally to all patients.

Only COVID-19 patients who have ever been prescribed nirmatrelvir/ritonavir during the study period were eligible for inclusion in the current study. We then excluded patients with missing index dates, who were aged <18 years at the index date, whose recorded date of nirmatrelvir/ritonavir prescription was before their index date, who were dead on or before the index date, or who had ever been prescribed molnupiravir before the index date or during the follow-up period (as oral antiviral use may impact VBR). Patients with contraindications to nirmatrelvir/ritonavir were also excluded, namely those with severe renal (eGFR <30 mL/min/1.73 m², dialysis, or renal transplantation) or liver disease (cirrhosis, hepatocellular carcinoma, or liver transplantation) at baseline[37], or those with drug contraindications to the concomitant use of nirmatrelvir/ritonavir[38]. For the target trial emulation specific to all-cause mortality or hospitalization, we also excluded patients who were hospitalized on or before the index date, as most such patients would have been diagnosed with COVID-19 during their hospital stay for other reasons.

For the target trial emulation on the outcome of VBR, only patients with at least one Ct value measurement within 14 days prior to the index date (day −14 to day 0) were included for analysis. Patients with more than one Ct value measurement pre or post diagnosis were mostly patients hospitalized for causes other than COVID-19 and who had received multiple RT-qPCR tests during their stay. Repeated testing was not necessarily performed due to clinical deterioration. For those diagnosed with COVID-19, official guidelines pre-March 2022 mandated particular Ct value cutoffs prior to discharge. Although such guidelines were rescinded in March 2022, some hospitals may have retained regular RT-qPCR testing as an indicative criterion for discharge. As discussed above, RT-qPCR testing was also broadly deployed in hospitals and in the community to pre-emptively identify COVID-19 patients. Nonetheless, to evaluate in isolation the risk of VBR from early nirmatrelvir/ritonavir administration in a purely outpatient setting, we also performed a subgroup analysis of a small cohort (289 individuals) of outpatient subjects with multiple Ct value measurements obtained solely via community-based RT-qPCR testing (which were widely available in Hong Kong during the study period).

## Outcomes

For our analysis on the risk of mortality and hospitalization, the outcome was a composite outcome of 28-day all-cause mortality or all-cause hospitalization since the index date. For our analysis on VBR, the outcome was the 28-day incidence of VBR from the index date. Adapting the definition used in our prior study[7], we defined VBR as a reduction in Ct value (provided by the SARS-CoV-2 RT-qPCR assays) between two consecutive measurements larger or equal to 3, and such decrease was sustained in at least the immediately subsequent Ct measurement ($\Delta Ct = Ct\_[before] - Ct\_[after 1] \geq 3$ and $\Delta Ct = Ct\_[before] - Ct\_[after 2] \geq 3$). As a proxy of viral burden, RT-qPCR Ct values are inversely correlated with viral load, i.e., a lower Ct value implies a higher viral burden. A decrease in Ct value by approximately 3 units was considered a rough estimate of an eight-fold increase in viral RNA. When the PCR result was negative, Ct value was not available and was imputed with value of 40, which was treated as the detection limit of the assay[39].

## Baseline covariates

Baseline characteristics of COVID-19 patients included age, sex, residential district (Hong Kong Island, Kowloon, New Territories, and others—the former two districts are entirely urban, while the latter two comprise of large urban or suburban settlements surrounded by rural areas), CCI, symptomatic presentation, concomitant use of corticosteroids, immunocompromised state (primary immunodeficiencies or on active immunosuppressive treatment at baseline or in the past year), healthcare utilization (inpatient and/or outpatient encounters) in the past year, previous SARS-CoV-2 infection, COVID-19 vaccination status, date of SARS-CoV-2 infection diagnosis and type of viral test for case detection (RAT or RT-qPCR). Being fully vaccinated against COVID-19 was defined as having received two doses of BNT162b2 (Comirnaty) or three doses of COVID-19 Vaccine (Vero Cell), Inactivated (CoronaVac); while boosted was defined as having received at least three doses of BNT162b2 or four doses of CoronaVac[40]. Type of care received by patients (i.e., inpatient or outpatient care) were included as additional covariates for VBR analysis.

## Statistical analysis

For each target trial emulation, we applied inverse probability weighting, cloning, censoring and inverse probability censoring-weighting (IPW-Cloning-Censoring-IPCW)[5,14,28] sequentially to estimate the absolute risk reduction (ARR) and relative risk (RR) of all-cause hospitalization/mortality and the ARR of VBR due to early

initiation of nirmatrelvir/ritonavir. We performed the same analytical workflow for our primary analysis with early initiation defined as days 0–1 and late initiation defined as days 2 or after, our sensitivity analyses with early initiation defined as days 0–2 or days 0–3 and late initiation defined as days 3 or after or days 4 or after, respectively, and all our other sensitivity and subgroup analyses.

We first compared the aforementioned baseline covariates between patients assigned to the early and late initiation groups. To ensure comparability between the early and late initiation groups and successful emulation of random assignment of strategies at baseline[5,28], we performed IPW to address any imbalance in baseline covariates between the two groups. Specifically, we constructed logistic regression models with the timing of nirmatrelvir/ritonavir initiation as the outcome conditional on all of the aforementioned baseline covariates, and estimated stabilized IPW for each patient. We then measured standardized mean differences (SMDs) between the two weighted groups for each covariate. A SMD of <0.1 implied sufficient balance of the baseline covariates between the weighted initiation groups and exchangeability thereof. Baseline characteristics before and after IPW, together with SMDs, for our primary analysis (i.e., early initiation defined as days 0–1, late initiation days 2–5) in both emulated target trials were provided in Table 1.

Using the weighted population groups in each emulated target trial, we then created two exact copies (clones) of all eligible patients and assigned one clone each to the early group or late group from the index date (time zero) until day 5. Clones were then censored on the day they deviated from the treatment protocol of their assigned group, i.e., in our primary analysis (early initiation defined as days 0–1), clones who received nirmatrelvir/ritonavir within 1 day of the index date were censored at day 1 in the late group, while clones who did not receive nirmatrelvir/ritonavir within 1 day were censored at day 1 in the early group. This cloning and censoring approach minimizes immortal time bias stemming from the lag between index date (when patients would have become eligible for nirmatrelvir/ritonavir treatment) and nirmatrelvir/ritonavir initiation (when patients would have been assigned to one of the two initiation groups)[14,32]. Subsequently, we applied inverse probability of censoring-weighting (IPCW)[5,41] on each cloned initiation group to address selection bias over time introduced by this informative censoring. A Cox regression model using baseline covariates described above was used to predict the probability that a clone would remain uncensored on each day, and to construct the corresponding inverse probability censoring weights to be applied to the clone. After IPCW, we applied a weighted nonparametric Kaplan–Meier estimator to estimate the ARR in each respective outcome (hospitalization/mortality or VBR) by 28 days due to early outpatient nirmatrelvir/ritonavir use, and RR at 28 days (i.e., ratio of cumulative incidence at 28 days of early group over that of late group), with 95% confidence intervals (CIs) constructed using a nonparametric bootstrap of 100 samples.

Subgroup analyses for both emulated trials were conducted for sex (male versus female), vaccination status (not fully vaccinated versus fully vaccinated/boosted), pre-existing comorbidities (CCI 0–6 versus >6), concomitant corticosteroid use (yes versus no), immunodeficiency status (immunocompromised versus not immunocompromised), secular time (March–June 2022, July–October 2022, and November 2022–January 2023), and availability of symptom onset date as index date (date of symptom onset was used as index date for 68,774 patients for mortality/hospitalization analysis and 1042 patients for VBR analysis, while date of diagnosis was used as index date for 18,404 patients for mortality/hospitalization analysis and 9072 patients for VBR analysis). For the study outcome of 28-day all-cause mortality or hospitalization, additional subgroup analysis was performed to differentiate patients with or without any Ct value measurements. We followed the same analytical workflow for each subgroup analysis as for our primary analysis, although we excluded

the subgroup in question from the list of baseline covariates (where applicable) when calculating IPWs for each initiation group. As mentioned above, we performed sensitivity analyses defining early initiation as initiation between 0 and 2 days or 0 and 3 days, and late initiation as initiation 3 days or after or 4 days or after respectively, for the hospitalization/mortality outcome overall, and for the VBR outcome overall and for all subgroups. In addition, we also performed further sensitivity analyses for mortality or hospitalization outcomes overall by (1) extending the time frame of posttreatment follow-up from 28 days in our main analysis to 42 days; and (2) excluding those who initiated nirmatrelvir/ritonavir beyond 5 days from the late initiators group.

The statistical analyses were performed using Stata version 18 (StataCorp LP, College Station, TX). All significance tests were two-tailed, where $P$ value < 0.05 was considered statistically significant.

### Ethics approval
This study was approved by the institutional review board of the University of Hong Kong/Hospital Authority Hong Kong West Cluster (reference no. UW 20-493). Individual patient-informed consent was not required for this retrospective cohort study using anonymized data.

### Reporting summary
Further information on research design is available in the Nature Portfolio Reporting Summary linked to this article.

## Data availability
The clinical outcome data and vaccination records were extracted from the Hospital Authority database in Hong Kong and data on confirmed cases of SARS-CoV-2 infection were extracted from the eSARS data provided by the Centre for Health Protection (Department of Health, The Government of the Hong Kong Special Administrative Region). The data custodians (the Hospital Authority and the Department of Health) provided the underlying individual patient data to The University of Hong Kong for the purpose of performing scientific research for the study. Restrictions apply to the availability of these data, which were used under licence of the Hospital Authority and the Department of Health for this study. The authors cannot transmit or release the data, in whole or in part in whatever form or media, or to any other parties or place outside Hong Kong; and the authors fully comply with the duties under the laws of Hong Kong relating to the protection of personal data including those under the Personal Data (Privacy) Ordinance and its principles in all aspects. For further information regarding the data request and approval process, please see the website of the Hospital Authority for the provision of data for research: https://www3.ha.org.hk/data/Provision/Submission. Hospital Authority data access inquiries can be directed to hacpaaedr@ha.org.hk.

## Code availability
The code used for this study is publicly available on GitHub (https://github.com/IvanAuHKU/COVID-nirmatrelvir-ritonavir-timing).

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

## Acknowledgements

This study was supported by the Health and Medical Research Fund (reference number: COVID190210 and CID-HKU2), Health Bureau,

Government of Hong Kong Special Administrative Region, China; and the AIR@InnoHK administered by Innovation and Technology Commission, Government of the Hong Kong Special Administrative Region, China. J.J.L. is supported by the HKU Presidential PhD Scholarship.

## Author contributions

J.J.L., C.K.H.W., and J.T.W. contributed to conceptualization, literature review, result interpretation, and manuscript writing. C.K.H.W. and I.C.H.A. contributed to data analysis. K.T.K.L. reviewed the literature, and contributed to the result interpretation. I.F.N.H., M.P., and G.M.L. contributed to the result interpretation, reviewed, and revised the manuscript.

## Competing interests

C.K.H.W. reports the receipt of the General Research Fund, Research Grant Council, Government of Hong Kong SAR; EuroQol Research Foundation; AstraZeneca and Boehringer Ingelheim, all outside the submitted work. I.F.N.H. is an advisory board member for Pfizer, MSD, AstraZeneca, Moderna, and GSK. The remaining authors declare no competing interests.
