## [Peer Review File · Nature Communications]

Optimal timing of nirmatrelvir/ritonavir treatment after COVID-19 symptom onset or diagnosis: a target trial emulationREVIEWER COMMENTS

Reviewer #1 (Remarks to the Author):

A key barrier to utilization of nirmatrelvir/ritonavir to prevent severe outcomes of Covid has been concern for 'rebound syndrome'. One strategy suggested by commentators and some clinicians, is to delay initiation by a few days with the idea that this would permit a more robust immune response preventing rebound. No evidence has been forwarded support this strategy and United States FDA review of nirmatrelvir/ritonavir found no evidence support hypothesis that treatment affected risk of rebound syndrome. However, additional analyses such as done by the authors remains important to confirm these assessments. The authors utilize routinely collected clinical data from Hong Kong two compare incidence of severe outcomes and, separately, development of rebound syndrome between two groups: early initiators (0-1 days) and later initiators (2-5 days). They have adopted some of the analytic methods used to compare nirmatrelvir/ritonavir versus no treatment, but this framework is likely insufficient to control expected bias. The findings are important but the large limitations decrease the significance of the findings. Given the complexities, would strongly suggest investigators analyze as target-trial emulation (see Hernan M NEJM 385:15, 2021).

Major:

1) Exchangeability. The exposure groups are quite different, particularly in regards to vaccination status and era of infection (consequently subvariant and likelihood of prior infection). The analytic approach attempts to control for factor associated with severe outcomes (eg, age, comorbidities, vaccination), but not factors that may explain early vs late treatment (eg, symptom severity, proximity to clinic, diagnosis of weekend, clinical progression, social disadvantage). I would assume that patients (or the clinicians caring for them) with progressive symptoms would seek treatment even if previously felt not needed on day 0 or 1. I don't think that the analytic design can adequately mitigate this bias that would favor prompt treatment group (includes both patients who would have had mild illness and those with progressive symptoms) over deferred treatment (enriched for those with progressive symptoms).

2) Asynchronous risk periods. The period of risk for patients treated promptly is from diagnosis/symptom day 0-1 to day 28-29, but for patients with deferred treatment it is day

2-5 to day 30-33. With risk substantially higher in the early period, these differences likely are consequential in a Cox model. Design with standardized definition (ideally symptom onset date) or doesn't rely on failure time would be less prone to bias.

3) Superiority or non-inferiority, primary rebound endpoint? The hypothesis offered is that delayed treatment would not reduce incidence of rebound syndrome. This hypothesis would typically be assessed using a non-inferiority design, however a superiority design was used. The incidence of rebound syndrome appears to be significantly lower for delayed nirmatrelvir/ritonavir as 95% confidence limits does not include the observed incidence for immediate treatment. The Cox model was not significant, but unsure if that was the primary endpoint. Additionally, the Cox model is vulnerable for bias due to variable time of remeasurement of CT value.

4) Rebound definition excludes symptoms. The primary concern for rebound syndrome is that it prolongs period of illness risk and potentially severe outcomes. The definition of rebound only consider viral load and does not include any information about symptoms.

5) Rebound ascertainment, measurement bias? Suggest authors provide further information about standard practice on follow-up viral measurement to frame why some patients had multiple measurements and others did not.

Minor

1) Survival curves. Suggest including survival curves for both severe outcomes and rebound.

2) Provide more contextual description. What were treatment guidelines in Hong Kong during this period that directed who was tested and treated? What proportion of increased risk individuals were tested? Were patients and providers counseled on rebound syndrome and did this guide treatment decisions.

3) Outside review. If a governmental/regulatory body reviewed manuscript prior to submission, please describe and outline what edits were made (if any).

Reviewer #2 (Remarks to the Author):

Wong et al. utilized EHR from Hong Kong to conduct a series of significant studies on COVID-19 vaccine and treatment effectiveness. The current study aimed to investigate the correlation between the timing of initiation of nirmatrelvir/ritonavir treatment and COVID-

19 outcomes. The results showed that patients who initiated treatment within one day of symptom onset or diagnosis had a significantly lower risk of mortality or hospitalization. I have several comments that hope would help further strengthen the study.

1. The lag time between symptom onset/diagnosis and the administration of treatment largely depends on factors such as testing policies, healthcare accessibility and treatment availability during the study period. The study enrolled participants between March 2022 and February 2023, which encompassed various waves of COVID-19 outbreaks and changes in public health policies in Hongkong. To facilitate comprehension by readers worldwide, the authors should provide a comprehensive description of the study setting in Hong Kong during the study period. This could include details such as the testing policy (e.g., whether mandatory testing was required), healthcare accessibility, treatment availability, and the method of administering the treatment.

Furthermore, it would be crucial to discuss how the differences in accessibility to testing, healthcare and treatment could result in varying lag times.

2. The reported HR of 0.48 (0.44, 0.53) on death or hospitalization for patients who received treatment within 0-1 days compared to those who received treatment between 2-5 days is surprising. Particularly, the magnitude of this difference appears to be even greater than the difference between those who received treatment with nirmatrelvir/ritonavir and those who did not, as reported by the same group use same data source (<https://pubmed.ncbi.nlm.nih.gov/36216007/>).

This raises the possibility that differences in some treatment or testing indications between groups may not have been accounted for. To address this, the authors may consider further adjustments of factors that may relate to the urgency of receiving treatment, such as specific comorbidities, symptoms at the time of diagnosis, and the route of testing and treatment administration. Additionally, it may be helpful if the authors additionally compare both treatment groups with an infected control group that did not receive treatment.

3. Categorized patients into 0-1 and 2-5 days for the timing of nirmatrelvir/ritonavir treatment initiation raises some questions. It is unclear what the rationale for this categorization. Moreover, based on the study hypothesis, it is unclear whether the authors expected that treatment initiation on day 2 would yield better outcomes than on day 3, and

similarly 4 vs 5? Given these concerns, it may be more appropriate to treat the lag time between diagnosis and treatment initiation as a non-linear continuous function in addition to using these arbitrary cutoffs.

4. Virus rebound is an important issue. However, I am not sure if the Ct measurements in this study is suitable to answer the question. Based on Figure 1, it appears that only a very small percentage of the cohort had Ct measurements available, and the hospitalization/death rates for this group were 100 times higher (47-64%) than those without Ct measurements (0.5-0.8%). This raises concerns about severe selection bias for those who had Ct measurements available and whether the results accurately represent the population of general nirmatrelvir/ritonavir users.

It is unclear whether the Ct values after hospitalization (after event) were included in the analyses.

It would be helpful if the authors could consider using other ways to measure or proxy the rebound in the overall study population.

RESPONSES TO REVIEW COMMENTS (Manuscript #NCOMMS-23-16121A - Optimal timing of nirmatrelvir/ritonavir treatment for reducing severity after symptom onset or diagnosis of COVID-19: emulation of a randomized trial using electronic health records in Hong Kong)

Reviewer #1 (Remarks to the Author):

A key barrier to utilization of nirmatrelvir/ritonavir to prevent severe outcomes of Covid has been concern for ‘rebound syndrome’. One strategy suggested by commentators and some clinicians, is to delay initiation by a few days with the idea that this would permit a more robust immune response preventing rebound. No evidence has been forwarded support this strategy and United States FDA review of nirmatrelvir/ritonavir found no evidence support hypothesis that treatment affected risk of rebound syndrome. However, additional analyses such as done by the authors remains important to confirm these assessments. The authors utilize routinely collected clinical data from Hong Kong to compare incidence of severe outcomes and, separately, development of rebound syndrome between two groups: early initiators (0-1 days) and later initiators (2-5 days). They have adopted some of the analytic methods used to compare nirmatrelvir/ritonavir versus no treatment, but this framework is likely insufficient to control expected bias. The findings are important but the large limitations decrease the significance of the findings. Given the complexities, would strongly suggest investigators analyze as target-trial emulation (see Hernan M NEJM 385:15, 2021).

Response 1.1: Thank you for your helpful comments. Pursuant to your feedback, we have substantially revised our manuscript to implement a target trial emulation design using our available observational data^{1,2}. Specifically, the revised paper followed the approach to target trial emulation as described in Hernán et al^{1,3}. First, to ensure compatibility between the early and late initiation groups and successful emulation of the random assignment of strategies at baseline, we performed inverse probability weighting to address and minimise any imbalance (and thus confounding) in baseline covariates between the two groups. In addition to risk factors for severe disease and clinical deterioration already included in the prior version of our manuscript, we incorporated additional baseline covariates, namely residential district, symptomatic presentation, healthcare utilization (any inpatient and/or outpatient encounters) in the past year, and type of viral test for case detection (rapid antigen test [RAT] or RT-qPCR [PCR]), to attempt to capture as many potential confounding factors attributable to differential access to healthcare or pace of symptomatic deterioration between the two initiation groups as

possible. Subsequently, following Hernán et al¹⁻³, we cloned all weighted subjects into both initiation groups, and censored clones when they ceased to be consistent with their assigned groups. This procedure eliminated immortal time bias stemming from the time lag between treatment eligibility (i.e. index date) and treatment assignment (i.e. prescription date). Finally, we applied inverse probability censoring-weighting to eliminate selection bias generated from the informative censoring of clones. We have elaborated the concepts of target trial emulation under the Methods section (page 36 line 224-227). Specifications of the main components of the hypothetical target trials and our two emulated trials using observational data were also detailed in Extended Data Table 1 (on all-cause mortality or hospitalization) and Extended Data Table 2 (on VBR).

In summary, our re-analyses conducted two separate target trial emulations, demonstrating lower risk of 28-day all-cause mortality/hospitalization in early group (Table 2 – absolute risk reduction [ARR]: 0.28%, 95% CI (0.07%, 0.49%), relative risk [RR]: 0.87 (0.78, 0.97), and no significant difference in VBR risk between the two groups (Table 3 – ARR: -2.11% (-5.32%, 0.89%) RR: 1.35 (0.89, 2.82)). Overall results were broadly consistent with original results.

References:

1. Hernán MA, Robins JM, Using Big Data to Emulate a Target Trial When a Randomized Trial Is Not Available, *American Journal of Epidemiology*, 2016; 183(8):758–764
2. Hernán MA, Wang W, Leaf DE. Target Trial Emulation: A Framework for Causal Inference From Observational Data. *JAMA*. 2022;328(24):2446–2447
3. Hernán, M.A., Sauer, B.C., Hernández-Díaz, S., Platt, R. & Shrier, I. Specifying a target trial prevents immortal time bias and other self-inflicted injuries in observational analyses. *J Clin Epidemiol* **79**, 70-75 (2016).

Major:

1) Exchangability. The exposure groups are quite different, particularly in regards to vaccination status and era of infection (consequently subvariant and likelihood of prior infection). The analytic approach attempts to control for factor associated with severe outcomes (eg, age, comorbidities, vaccination), but not factors that may explain early vs late treatment (eg, symptom severity, proximity to clinic, diagnosis of weekend, clinical progression, social disadvantage). I would assume that patients (or the clinicians caring for them) with progressive symptoms would seek treatment even if previously felt not

needed on day 0 or 1. I don't think that the analytic design can adequately mitigate this bias that would favor prompt treatment group (includes both patients who would have had mild illness and those with progressive symptoms) over deferred treatment (enriched for those with progressive symptoms).

Response 1.2: Thank you for your helpful comments. As described in Response 1.1, in addition to potential factors associated with severe outcomes, we have now added several additional baseline covariates of COVID-19 patients to address differences in patient symptomatology and access to healthcare in our updated inverse probability weighting model. We then applied inverse probability weights to balance all baseline covariates between the early and late treatment groups to maximize comparability (or exchangeability) between the two cohorts. We submit that our expanded array of baseline covariates allows for adequate balancing of the risks of clinical deterioration and differences in the delay in seeking medical care between the early and late treatment groups in both our target trial emulations.

Throughout the study period, residents had free access to PCR tests via community testing centres (available across the living areas and districts in Hong Kong); or alternatively, they could self-arrange RAT and report positive results to the Department of Health. Prescription of oral antivirals and disease management of COVID-19 patients were available for free for all Hong Kong residents from public healthcare facilities (hospitals and outpatient designated COVID-19 clinics) managed by the Hospital Authority, the statutory body managing all public healthcare services in Hong Kong (page 31 line 95-110). Public healthcare facilities are located in all residential districts in Hong Kong and many were open seven days a week and during all public holidays during our study period. The wide availability of healthcare services is exemplified by the roughly equal distribution of weekday and weekend COVID-19 diagnoses between early and late treatment groups in both of our target trial emulations. (Table 1).

While we acknowledge that clinical progression of symptoms might have an impact on the healthcare seeking behaviour of patients and hence the timing of antiviral initiation; unfortunately, our data sources do not contain detailed information on the progressive severity of symptoms (if any) of individual patients, and we could only adjust for any symptomatic presentation of patients at the point of physician-patient contact (this included recorded date of symptom onset or self-report of any COVID-19-related symptoms upon SARS-CoV-2 infection diagnosis). Further, we submit that, given that nirmatrelvir/ritonavir-eligible patients

were already of high risk (i.e. ≥ 60 years old, immunocompromised, etc), those experiencing clinical deterioration and thus presenting themselves to public healthcare facilities (often to Accident/Emergency) would have most likely been hospitalised immediately or soon after and be prescribed nirmatrelvir/ritonavir in an inpatient setting. Such patients would thus have been entirely excluded from our hospitalization/mortality study given our exclusion of patients who were hospitalized/died within 2 days of index date. In effect, a bias that would favour the prompt treatment group over the deferred treatment group due to enrichment of the latter with patients experiencing clinical deterioration would have been unlikely to occur in our updated analysis. The above discussion has now been incorporated into the Limitations section of our manuscript (page 14 line 318-332)

2) Asynchronous risk periods. The period of risk for patients treated promptly is from diagnosis/symptom day 0-1 to day 28-29, but for patients with deferred treatment it is day 2-5 to day 30-33. With risk substantially higher in the early period, these differences likely are consequential in a Cox model. Design with standardized definition (ideally symptom onset date) or doesn't rely on failure time would be less prone to bias.

Response 1.3: In this revised manuscript, patient inclusion period was revised to between 16th March 2022 and 15th January 2023, allowing all patients to have at least 28-days of follow-up for the main analyses before the administrative end of observation period, i.e. 12th February 2023. Risk periods for patients in both early and late initiation groups were at a maximum of 28 days (median of 28 days for both groups). We now set the earlier date of SARS-CoV-2 infection diagnosis or symptom onset as the index date (or time zero) for all patients, and followed them up to 28 days (page 32 line 135-138). Hence, the risk periods for both groups were no longer asynchronous in this version.

3) Superiority or non-inferiority, primary rebound endpoint? The hypothesis offered is that delayed treatment would not reduce incidence of rebound syndrome. This hypothesis would typically be assessed using a non-inferiority design, however a superiority design was used. The incidence of rebound syndrome appears to be significantly lower for delayed nirmatrelvir/ritonavir as 95% confidence limits does not include the observed incidence for immediate treatment. The Cox model was not significant, but unsure if that was the primary endpoint. Additionally, the Cox model is vulnerable for bias due to variable time of remeasurement of CT value.

Response 1.4: Although we originally hypothesized that initiating nirmatrelvir/ritonavir treatment immediately after the symptom onset or diagnosis of COVID-19 would not have elevated the risk of VBR when compared to delayed initiation, the hypotheses stated in our original submission was not the null hypothesis subject to the statistical testing. The primary aim of this study was to estimate the ARR and RR of all-cause hospitalization/mortality or VBR due to early initiation of nirmatrelvir/ritonavir, and to further assess the effect of the timing of treatment on the risk of VBR bi-directionally, as it remains unclear whether and by how much an earlier suppression of viral load would increase or decrease the risk of VBR. Superiority or non-inferiority analyses are by definition unidirectional and thus would not have permitted such a bidirectional analysis. Correspondingly, we have also clarified our manuscript, specifying our study objective as that of measuring the ARR and RR of hospitalization/death or VBR from early versus late initiation of nirmatrelvir/ritonavir (page 5 lines 88-91), rather than testing any particular *a priori* hypothesis. Further, we have now conducted our measurement of the ARR and RR from early initiation using a non-parametric Kaplan-Meier estimator (as hazards would likely have been not proportional due to the cloning steps in our revised target trial emulation study design⁴). The K-M estimator is agnostic to the variable time of remeasurement of Ct values.

Reference:

4 Maringe C, Benitez Majano S, Exarchakou A, Smith M, Rachet B, Belot A, Leyrat C. Reflection on modern methods: trial emulation in the presence of immortal-time bias. Assessing the benefit of major surgery for elderly lung cancer patients using observational data. *Int J Epidemiol.* 2020 Oct 1;49(5):1719-1729. doi: 10.1093/ije/dyaa057. PMID: 32386426; PMCID: PMC7823243.

4) Rebound definition excludes symptoms. The primary concern for rebound syndrome is that it prolongs period of illness risk and potentially severe outcomes. The definition of rebound only consider viral load and does not include any information about symptoms.

Response 1.5: Unfortunately, our dataset does not contain detailed records of the recurrence and duration of symptoms after the index date, restricting us from assessing rebound from a symptomatology perspective. We have thus noted this as a limitation in our revised manuscript (page 14 lines 318-321). Nonetheless, while we recognise the importance of identifying symptomatic rebound, we respectfully submit that our definition of viral burden rebound by Ct value measurements (specifically, an 8-fold increase in viral load post treatment) provides a

more standardized definition of the post-treatment rebound phenomena than symptomatic rebound and alleviates the need to rely on subjective patient self-reports and clinician judgment. Further, our definition may also potentially capture a larger subset of rebound patients by including those who were asymptomatic or experienced extremely mild symptoms during their rebound phase (but who may have remained infectious throughout).

5) Rebound ascertainment, measurement bias? Suggest authors provide further information about standard practice on follow-up viral measurement to frame why some patients had multiple measurements and others did not.

Response 1.6: At the beginning of the outbreak of COVID-19 in Hong Kong, while some patients were initially diagnosed by rapid antigen test (RAT), virtually everyone would undergo PCR testing on admission as per protocol and indeed retested by RT-PCR from time to time after admission to monitor viral load evolution. For monitoring during treatment according to clinical management guidelines¹, patients were subject to repeated nasopharyngeal flocked swab (NPS), nasopharyngeal aspirate, sputum, deep throat saliva or stool specimens of RT-PCR testing twice 24 hours apart before release from isolation. However, there were no official recommendations on how frequently RT-PCR tests should be performed to monitor the changes in viral burden among inpatients. Repeated testing was not necessarily performed due to clinical deterioration. In clinical practice, nasal and/or throat swabs were collected for routine testing in hospital settings, while deep throat saliva was reserved for screening or routine testing in the community. Stool specimens were not routinely collected, unless clinically indicated for individual patients. Depending on the healthcare and human resources available, different hospitals might adopt different practices to monitor the changes in viral burden among inpatients; for instance, PCR tests by NPS could be repeated on alternate days in medically stable patients, while daily PCR tests were conducted for those with severe COVID-19, those who were immunocompromised, or those receiving intensive care.

Nevertheless, at the start of our study period (March 2022), the public healthcare system in Hong Kong was overwhelmed with newly diagnosed COVID-19 cases during the wave peak predominated by Omicron BA.2 infection. To facilitate the turnover of hospital beds, the Hospital Authority guidelines have since then been updated on the criteria for discharge of COVID-19 patients from hospital, where clinically stable patients could be discharged to home premises or residential care facilities to continue their isolation until negative viral test results were obtained². As particular Ct value cutoffs were no longer adopted as one of the essential

criteria for discharge, Ct value measurements might not have been performed as regularly; and for patients managed in the community setting, they were allowed to end their isolation or home quarantine upon negative RAT results for two consecutive days (PCR tests were not compulsory for these patients, and hence Ct value measurements might not be available). Nonetheless, some hospitals may have retained repeated PCR testing as an indicative tool for evaluating patient eligibility for discharge. Furthermore, our dataset contains a small subset of entirely outpatient COVID-19 patients who received multiple PCR tests in the community. We have separately analysed the risk of VBR amongst these patients and found the results to be largely equivalent to those who have received repeated PCR tests in the hospital. These are now elaborated under the Methods section (page 34 lines 180-192).

In summary, we have now addressed the potential ascertainment bias in the measurement of VBR under our study limitations as follows: “Besides, as Ct value measurements were mainly collected from hospitalized patients, and testing frequency might have depended on patient severity, physician clinical judgment and the availability of healthcare resources; ascertainment bias in the VBR study outcome could not be ruled out.” (page 13 line 307-310).

References:

1. HA Central Committee on Infectious Diseases and Emergency Response (CCIDER). Interim Recommendation on Clinical Management of Adult Cases with Coronavirus Disease 2019 (COVID-19). Version 1.12. (2022).
2. The Government of the Hong Kong Special Administrative Region. Government announces latest criteria for discharge from isolation and home quarantine, <<https://www.info.gov.hk/gia/general/202202/26/P2022022600750.htm>> (2022).

Minor

1) Survival curves. Suggest including survival curves for both severe outcomes and rebound.

Response 1.7: We have added the cumulative incidence plots for outcomes of 1) 28-day all-cause mortality or all-cause hospitalization, and 2) viral burden rebound to Extended Data Figures 1 and 3, respectively.

2) Provide more contextual description. What were treatment guidelines in Hong Kong during this period that directed who was tested and treated? What proportion of increased risk individuals were tested? Were patients and providers counseled on rebound syndrome and did this guide treatment decisions.

Response 1.8: According to clinical management guidelines for COVID-19 from the Hong Kong Hospital Authority (HA)¹, patients who 1) had mild symptoms, 2) were at risk of progressing to severe disease (i.e. diabetes mellitus, obesity with BMI of $\geq 30\text{kg/m}^2$, age ≥ 60 years, immunocompromised state, underlying chronic illnesses, or not fully vaccinated), and 3) at early stage of disease (within five days of symptom onset) were recommended to receive molnupiravir or nirmatrelvir/ritonavir. Later versions (since 21st March 2022) of the guidelines also added that nirmatrelvir/ritonavir should be preferentially administered over molnupiravir, unless the patient was on any concomitant medication(s) contraindicated for nirmatrelvir/ritonavir. We have added a summary of the nirmatrelvir/ritonavir use during the study period under the Methods section (page 32 lines 147-155).

As advocated by the Hong Kong government, individuals with COVID-19-related symptoms or recent contact with infected patients were encouraged to be tested; and for high-risk groups such as healthcare workers and staff at residential care facilities, regular mandatory testing was required. Up to late 2022, compulsory testing notices were also issued to residents of residential premises visited by tested positive cases or where sewage samples indicated the presence of SARS-CoV-2 viral RNA. Throughout the study period, residents had free access to PCR tests (for voluntary or compulsory testing) via community testing centres (available across all residential districts in Hong Kong); or alternatively, they could self-arrange RAT (which were widely and freely distributed by the Government) and report positive results to the Department of Health. Prescription of oral antivirals and disease management of COVID-19 patients were available from all public healthcare facilities (hospitals and outpatient clinics) managed by the HA, the statutory body managing all public healthcare services in Hong Kong (page 31 lines 97-110).

Unfortunately, we do not have information on whether counselling on rebound syndrome was provided by medical professionals during the consultation process, or whether the treatment decisions were altered by the process.

Reference:

1. HA Central Committee on Infectious Diseases and Emergency Response (CCIDER). Interim Recommendation on Clinical Management of Adult Cases with Coronavirus Disease 2019 (COVID-19). Version 1.12. (2022).

3) Outside review. If a governmental/regulatory body reviewed manuscript prior to submission, please describe and outline what edits were made (if any).

Response 1.9: The manuscript has not been reviewed by a government or regulatory body prior to submission.

Reviewer #2 (Remarks to the Author):

Wong et al. utilized EHR from Hong Kong to conduct a series of significant studies on COVID-19 vaccine and treatment effectiveness. The current study aimed to investigate the correlation between the timing of initiation of nirmatrelvir/ritonavir treatment and COVID19 outcomes. The results showed that patients who initiated treatment within one day of symptom onset or diagnosis had a significantly lower risk of mortality or hospitalization. I have several comments that hope would help further strengthen the study.

1. The lag time between symptom onset/diagnosis and the administration of treatment largely depends on factors such as testing policies, healthcare accessibility and treatment availability during the study period. The study enrolled participants between March 2022 and February 2023, which encompassed various waves of COVID-19 outbreaks and changes in public health policies in Hongkong. To facilitate comprehension by readers worldwide, the authors should provide a comprehensive description of the study setting in Hong Kong during the study period. This could include details such as the testing policy (e.g., whether mandatory testing was required), healthcare accessibility, treatment availability, and the method of administering the treatment. Furthermore, it would be crucial to discuss how the differences in accessibility to testing, healthcare and treatment could result in varying lag times.

Response 2.1: We acknowledge that the lag time between the index date (earlier date of diagnosis or symptom onset) and nirmatrelvir/ritonavir initiation could be affected by factors such as testing policies, healthcare accessibility and treatment availability during the study period. As such, we have provided the following additional context on our manuscript.

As advocated by the Hong Kong government, individuals with COVID-19-related symptoms or recent contact with infected patients were encouraged to be tested; and for high-risk groups such as healthcare workers and staff at residential care facilities, regular mandatory testing was required. Up to late 2022, compulsory testing notices were also regularly issued to residents of residential premises visited by tested positive cases or where sewage samples indicated the presence of SARS-CoV-2 viral RNA. Residents had free access to RT-qPCR [PCR] tests via community testing centres (available across the living areas and districts in Hong Kong); or alternatively, they could self-arrange RAT (which were widely and freely distributed by the Government) and report positive results to the Department of Health (page 31 lines 97-105).

During our study period, prescription of oral antivirals and disease management of COVID-19 patients were available from all public healthcare facilities (hospitals and outpatient clinics) managed by the Hospital Authority (HA), the statutory body managing all public healthcare services in Hong Kong (page 30 lines 105-108). Eligible patients were usually prescribed nirmatrelvir/ritonavir (preferred) or molnupiravir (if contraindicated to nirmatrelvir/ritonavir) upon diagnosis at the point of care without further delay¹. Since April 2022, the Department of Health had provided private doctors with oral antiviral medications (both molnupiravir and nirmatrelvir/ritonavir) for prescription to COVID-19 patients seeking private medical care². Private doctors were asked to follow the HA treatment guidelines in the eligibility assessment and prioritisation of oral antivirals to high-risk patients; however, information of COVID-19 patients managed in the private healthcare system were not available from our data sources.

At the start of our study period (March 2022), the public healthcare system in Hong Kong was overwhelmed with newly diagnosed COVID-19 cases during the wave peak predominated by Omicron BA.2 infection. Potential delays in the confirmation of cases by PCR tests and patients attending medical visits for oral antiviral prescription could not be ruled out. Such potential increase in the lag time may explain the heterogeneity of treatment effects during the period from March to June 2022. We have now discussed how the accessibility to testing, healthcare and treatment could have impacted lag times, and hence the results of our secular time subgroup analyses in the Discussion section of our revised manuscript (page 13 line 291-304). We also performed a separate subgroup analysis segregating patients by how their index dates were defined (symptom onset vs diagnosis). Results in both our hospitalization/mortality and VBR analyses were similar across patients in both index date definition subgroups.

Nirmatrelvir/ritonavir was administered orally for all treated patients in the current study according to their drug prescription and dispensing records. This has now been added to the Methods section as follows: “Nirmatrelvir/ritonavir treatment was administered orally to all patients.” (page 33 line 159)

References:

1. HA Central Committee on Infectious Diseases and Emergency Response (CCIDER). Interim Recommendation on Clinical Management of Adult Cases with Coronavirus Disease 2019 (COVID-19). Version 1.12. (2022).
2. The Government of the Hong Kong Special Administrative Region. List of private doctors who can prescribe COVID-19 oral drugs published, <<https://www.info.gov.hk/gia/general/202204/26/P2022042600552.htm>> (2022).

2. The reported HR of 0.48 (0.44, 0.53) on death or hospitalization for patients who received treatment within 0-1 days compared to those who received treatment between 2-5 days is surprising. Particularly, the magnitude of this difference appears to be even greater than the difference between those who received treatment with nirmatrelvir/ritonavir and those who did not, as reported by the same group use same data source (<https://pubmed.ncbi.nlm.nih.gov/36216007/>). This raises the possibility that differences in some treatment or testing indications between groups may not have been accounted for. To address this, the authors may consider further adjustments of factors that may relate to the urgency of receiving treatment, such as specific comorbidities, symptoms at the time of diagnosis, and the route of testing and treatment administration. Additionally, it may be helpful if the authors additionally compare both treatment groups with an infected control group that did not receive treatment.

Response 2.2: Pursuant to feedback from Reviewer #1, we have substantially revised our manuscript to adopt a target trial emulation approach using our available observational data. Specifically, following Hernan et al¹⁻³, we sequentially applied an inverse probability weighting (IPW) – cloning – censoring and inverse probability censoring-weighting (IPCW) workflow to our primary analysis for hospitalization/mortality and VBR, and separately to all our sensitivity and subgroup analyses. The target trial emulation approach provides a structured framework to ensure comparability and exchangeability between our two treatment groups, via adjustment of confounders in the IPW step. Via cloning-censoring-IPCW, it also minimizes

the effect of immortal time bias stemming from the time lag between treatment eligibility (i.e. at index date) and the assignment of treatment (i.e. whether patients were eventually prescribed treatment early or late). (page 36 lines 224-236; page 37 lines 252-258).

Exchangeability between the early and late initiation groups is a prerequisite to successful emulation of a target trial – specifically, the random assignment of strategies at baseline. In practice, this entails adjustment for potential confounders influencing both treatment assignment and outcome. Accordingly, we first balanced baseline characteristics between the early and late initiation groups using inverse probability weightings constructed via propensity scores. In addition to potential factors associated with severe COVID-19 outcomes (which were already included in our original manuscript), we have now added several baseline covariates of patients to address the risk of symptomatic deterioration and access to healthcare, namely residential district, any symptomatic presentation, healthcare utilization (any inpatient and/or outpatient encounters) in the past year, and type of viral test for case detection (RAT or RT-qPCR), in our updated inverse probability weightings model (page 5 line 108-109). Unfortunately, our data sources do not contain detailed information on the severity of symptoms (if any) of individual patients at the point of physician-patient contact. Nonetheless, we submit that our expanded array of baseline covariates already allows for a comprehensive adjustment of factors related to the urgency of receiving treatment.

Following the target trial emulation approach, its adjustment for additional baseline covariates, and the subsequent steps of cloning-censoring-IPCW, our updated analyses on the study outcome of 28-day all-cause mortality or hospitalization continued to suggest a lower risk for early versus late initiation of nirmatrelvir/ritonavir consistent with our results in the previous version of the manuscript (adjusted HR=0.48, 95%CI=0.44-0.53), albeit with a much reduced magnitude of effect (absolute risk reduction: 0.28%, 95%CI (0.07%, 0.49%), relative risk: 0.87 (0.78,0.97)). Please note that, in our latest revision, we have replaced the hazard rate with absolute risk reduction (ARR) and relative risk (RR), as the target trial emulation design generates unproportional hazards not suitable for Cox regression, necessitating the switch to a non-parametric Kaplan-Meier estimator⁴.

Lastly, we respectfully submit that the population of untreated but diagnosed COVID-19 patients in Hong Kong does not provide a suitable control group. After the wide availability of nirmatrelvir/ritonavir beginning in May-June 2022, all eligible patients (e.g. all above 60 years

old, or the immunocompromised) were immediately prescribed nirmatrelvir/ritonavir (preferred) or molnupiravir upon diagnosis. Those not eligible and not prescribed antiviral medication were uniformly younger and/or immunocompetent and thus faced a significantly lower risk of severe disease and death. As a result, they would not provide an appropriate control group for analysis.

References:

1. Hernán MA, Robins JM, Using Big Data to Emulate a Target Trial When a Randomized Trial Is Not Available, *American Journal of Epidemiology*, 2016; 183(8):758–764
2. Hernán MA, Wang W, Leaf DE. Target Trial Emulation: A Framework for Causal Inference From Observational Data. *JAMA*. 2022;328(24):2446–2447
3. Hernán, M.A., Sauer, B.C., Hernández-Díaz, S., Platt, R. & Shrier, I. Specifying a target trial prevents immortal time bias and other self-inflicted injuries in observational analyses. *J Clin Epidemiol* 79, 70-75 (2016).
- 4 Maringe C, Benitez Majano S, Exarchakou A, Smith M, Rachet B, Belot A, Leyrat C. Reflection on modern methods: trial emulation in the presence of immortal-time bias. Assessing the benefit of major surgery for elderly lung cancer patients using observational data. *Int J Epidemiol*. 2020 Oct 1;49(5):1719-1729. doi: 10.1093/ije/dyaa057. PMID: 32386426; PMCID: PMC7823243.

3. Categorized patients into 0-1 and 2-5 days for the timing of nirmatrelvir/ritonavir treatment initiation raises some questions. It is unclear what the rationale for this categorization. Moreover, based on the study hypothesis, it is unclear whether the authors expected that treatment initiation on day 2 would yield better outcomes than on day 3, and similarly 4 vs 5? Given these concerns, it may be more appropriate to treat the lag time between diagnosis and treatment initiation as a non-linear continuous function in addition to using these arbitrary cutoffs.

Response 2.3: Pursuant to feedback from Reviewer #1, we have adopted a target trial emulation strategy in the latest version of our manuscript. As of the time of writing, existing theoretical target trial emulation literature do not provide clear guidance on how treatment delay (i.e. time lag between diagnosis and treatment) could itself be incorporated as a continuous predictor without the presence of some other clinical indicator serving as a threshold for treatment initiation (for example, CD4 count thresholds at which combined antiretroviral therapy (cART) should be initiated^{1,2}). Since our current study did not include a clinical threshold for treatment

initiation (other than time itself), we submit that incorporating time as a “standalone” non-linear continuous predictor may not enjoy sufficient theoretical basis under the target trial emulation framework.

Nonetheless, we recognised that a Day 0-1 / 2-5 cutoff may be arbitrary and thus added additional sensitivity analyses (for both our hospitalization/mortality and VBR analyses) redefining early initiation as between Days 0-2 or Days 0-3, and late initiation as between Days 3-5 and 4-5 respectively. The results were presented on Tables 2 and 3. Our sensitivity analyses revealed that initiating nirmatrelvir/ritonavir within Days 0-2 continued to provide a statistically significant reduction in the risk of hospitalization/death, although such an effect was no longer statistically significant if nirmatrelvir/ritonavir was initiated within Days 0-3. Meanwhile, the risks of VBR amongst the male, fully vaccinated or boosted and immunocompromised subgroups were significant if early treatment was defined as Days 0-1, but were no longer significant if early treatment was defined as Days 0-2 or 0-3 (for males) or as Days 0-3 (for the fully vaccinated or boosted, and the immunocompromised).

References:

¹ Cain LE, Robins JM, Lanoy E, Logan R, Costagliola D, Hernán MA. When to start treatment? A systematic approach to the comparison of dynamic regimes using observational data. *Int J Biostat.* 2010;6(2):Article 18. doi: 10.2202/1557-4679.1212. PMID: 21972433; PMCID: PMC3406513.

² HIV-CAUSAL Collaboration; Cain LE, Logan R, Robins JM, Sterne JA, Sabin C, Bansi L, Justice A, Goulet J, van Sighem A, de Wolf F, Bucher HC, von Wyl V, Esteve A, Casabona J, del Amo J, Moreno S, Seng R, Meyer L, Perez-Hoyos S, Muga R, Lodi S, Lanoy E, Costagliola D, Hernan MA. When to initiate combined antiretroviral therapy to reduce mortality and AIDS-defining illness in HIV-infected persons in developed countries: an observational study. *Ann Intern Med.* 2011 Apr 19;154(8):509-15. doi: 10.7326/0003-4819-154-8-201104190-00001. PMID: 21502648; PMCID: PMC3610527.

4. Virus rebound is an important issue. However, I am not sure if the Ct measurements in this study is suitable to answer the question. Based on Figure 1, it appears that only a very small percentage of the cohort had Ct measurements available, and the hospitalization/death rates for this group were 100 times higher (47-64%) than those without Ct measurements (0.5-0.8%). This raises concerns about severe selection bias for

those who had Ct measurements available and whether the results accurately represent the population of general nirmatrelvir/ritonavir users. It is unclear whether the Ct values after hospitalization (after event) were included in the analyses. It would be helpful if the authors could consider using other ways to measure or proxy the rebound in the overall study population.

Response 2.4: We agreed with your observation on the low proportion of outpatients for VBR analysis, and potential selection bias of including only those who had two or more Ct value measurements for VBR analysis, as most of such tests were collected during inpatient stays (thus the higher risk of hospitalization/death amongst patients with Ct measurements available). Such selection bias has been acknowledged in the limitation of the revised manuscript (page 13 lines 304-307).

Nonetheless, we note that patients with more than one Ct value measurement were mostly patients hospitalized for causes other than COVID-19, and received multiple RT-qPCR tests during their stay. Repeated testing was not necessarily performed due to clinical deterioration (page 33 lines 182-185). Further, our dataset contains a small subject of entirely outpatient COVID-19 patients who received multiple PCR tests in the community (usually as a result of mandatory testing). None of these patients were hospitalized nor died during the 28-day window from their index dates. We separately analyzed the risk of VBR amongst these patients and found the results to be largely equivalent to those who have received repeated PCR tests in the hospital. This potentially indicates that any selection bias due to the over-abundance of hospitalised patients in our VBR analysis was likely limited.

Meanwhile, for our VBR analysis, patients were observed from the index date (time zero) until that of VBR (event outcome), 28 days after the index date, or the administrative end of the follow-up period (12th February 2023), whichever the earliest (Extended Data Table 2). Patients would only be censored at registered death (competing event) but not upon hospital admission. Ct value measurements between time zero and day 28, including those obtained during hospitalization, were included for the detection of VBR (page 32 lines 138-139; page 33 lines 140-141).

Finally, we have considered alternative ways to measure the rate of rebound but our dataset contains no information on the recurrence and duration of symptoms after the index date. This

has been added to the study limitations of our revised manuscript (page 14 line 318-321). Respectfully, we submit that our definition of viral burden rebound by Ct value measurements (specifically, an 8-fold increase in viral load post treatment) provides an objective definition of the post-treatment rebound phenomena and alleviates the need to rely on subjective patient self-reports and clinician judgment. In addition, our definition may also potentially capture a larger subset of rebound patients by including those who were asymptomatic or experienced extremely mild symptoms during their rebound phase (but who may have remained infectious throughout).

REVIEWER COMMENTS

Reviewer #1 (Remarks to the Author):

The authors made substantial changes in analytic approach to address bias. The revised manuscript is strong and carefully addresses (and acknowledges where it cannot) threats to causal interpretation of their findings. My initial concerns have been addressed and the question that the investigators remains clinically important.

Reviewer #2 (Remarks to the Author):

I appreciate the authors' huge efforts in reanalyzing the study using a more rigorous approach, incorporating additional variables, sensitivity analyses, and more. With regard to the newly applied emulation of the target trial, I have a few questions about if it has been correctly applied.

1. The authors mention the use of the clone method between days 0 and 5. However, the conditions under which a participant would be censored after day 1 in each group remain unclear. Based on their design, it seems no violation of protocol would occur after day 1.

If a participant experiences an event on day 1, would this be considered a violation of the late treatment protocol? Similarly, if an event occurs on the date of treatment initiation, would this also be seen as a violation of the protocol or an event?

2. In extended data figure 1, the CIF indicates no events in either group during the initial days, which seems implausible given the study protocol. How could the early treatment group shows no risk in the first couple days? This appears to be inconsistent with the principles of emulating a target trial.

3. The principle of designing a target trial and subsequently emulating it appears has not been 100% followed in this study. Specifically, the authors seem to treat status changes that occur after treatment initiation and after randomization as part of the eligibility criteria. In the study, they removed participants hospitalized on the date of nirmatrelvir initiation, the use of nirmatrelvir after day 5, and mandating a Ct value post-treatment and more

before the emulation of randomization. This is exactly what emulation of target trial hope help avoid from observational study.

These changes of status after the designed randomization should be viewed as censoring events at their occurrence dates. An evaluation should be undertaken to ensure that random censoring is achievable via the IPCW after accounting for these censoring.

4. While the authors demonstrate successful balance at baseline, it remains unclear whether the IPCW sufficiently emulated the censoring at random for the two emulated trials. To substantiate the validity of the VBR analyses within the emulation framework, it's important for the authors to establish the ability to emulate not only the randomization of early versus late treatment initiation but also the randomization of the presence or absence of a Ct value.

RESPONSES TO REVIEW COMMENTS (NCOMMS-23-16121)

We provide here our point-by-point responses to the Reviewers' comments to an earlier version of this paper reviewed. We have now updated our manuscript addressing Reviewer #2's feedback on the design of our target trial emulations. Our updated analyses found overall results that were broadly consistent with results from our earlier submission.

Reviewer #1 (Remarks to the Author):

The authors made substantial changes in analytic approach to address bias. The revised manuscript is strong and carefully addresses (and acknowledges where it cannot) threats to causal interpretation of their findings. My initial concerns have been addressed and the question that the investigators remains clinically important.

Response 1.1: Thank you for your helpful comments and support.

Reviewer #2 (Remarks to the Author):

I appreciate the authors' huge efforts in reanalyzing the study using a more rigorous approach, incorporating additional variables, sensitivity analyses, and more. With regard to the newly applied emulation of the target trial, I have a few questions about if it has been correctly applied.

1. The authors mention the use of the clone method between days 0 and 5. However, the conditions under which a participant would be censored after day 1 in each group remain unclear. Based on their design, it seems no violation of protocol would occur after day 1.

If a participant experiences an event on day 1, would this be considered a violation of the late treatment protocol? Similarly, if an event occurs on the date of treatment initiation, would this also be seen as a violation of the protocol or an event?

Response 2.1: Thank you for your helpful comments. In our target trial emulation, clones are censored on the first day they no longer conform with their assigned treatment protocol (e.g. if early initiation is defined as Days 0-1, late treatment group clones who received nirmatrelvir/ritonavir within one day of the index date would be censored at Day 1, and early treatment clones who did not receive nirmatrelvir/ritonavir within one day would be censored at Day 1).

In our revised approach with respect to the mortality/hospitalisation analysis, we no longer exclude (or censor) patients who experience an event on day 1 after the index date, nor those patients who received nirmatrelvir/ritonavir more than 5 days after the index date (page 32-33 line 163-174). Accordingly, a late group clone who did not receive nirmatrelvir/ritonavir within one day, and who experienced a hospitalisation/mortality event on day 1, would be counted towards an event occurrence, and would not have been considered as having violated the late treatment protocol. However, we have continued to exclude patients who experience a hospitalisation event on the index date (i.e. date of symptom onset or COVID-19 diagnosis) (page [33] line [171-175]). Although we do not have access to detailed patient clinical records, we note from clinical practice that most such patients would have been diagnosed with COVID-19 during their hospital stay for other reasons. Thus, it would have been inappropriate to treat such patients as having experienced a hospitalisation event.

Given our definition of a VBR event, which requires consecutive post-treatment Ct-value measurements across multiple RT-qPCR assays, no VBR events have occurred within Days 0-1 from the index date.

2. In extended data figure 1, the CIF indicates no events in either group during the initial days, which seems implausible given the study protocol. How could the early treatment group shows no risk in the first couple days? This appears to be inconsistent with the principles of emulating a target trial.

Response 2.2: Under our revised protocol, the current version of our Extended Data Figure 1 shows that both the early and late treatment groups have experienced mortality/hospitalisation

events in the first couple days after the index date.

3. The principle of designing a target trial and subsequently emulating it appears has not been 100% followed in this study. Specifically, the authors seem to treat status changes that occur after treatment initiation and after randomization as part of the eligibility criteria. In the study, they removed participants hospitalized on the date of nirmatrelvir initiation, the use of nirmatrelvir after day 5, and mandating a Ct value post-treatment and more before the emulation of randomization. This is exactly what emulation of target trial hope help avoid from observational study.

These changes of status after the designed randomization should be viewed as censoring events at their occurrence dates. An evaluation should be undertaken to ensure that random censoring is achievable via the IPCW after accounting for these censoring.

Response 2.3: In designing our revised approach, we have ensured that no post-index date information would be used for either assigning patients and clones into treatment arms nor including/excluding patients into the study as a whole. As such, in our mortality/hospitalisation analysis, we have now added back all patients who were hospitalised/died within 2 days of nirmatrelvir/ritonavir initiation, and all patients who initiated nirmatrelvir/ritonavir after day 5. For reasons discussed above, we excluded patients who were hospitalized on or before the index date (please note that the index date would, in most cases, differ from the date of nirmatrelvir/ritonavir initiation).

Meanwhile, in order to remove any use of post-index date information (i.e. availability of Ct-value post treatment) for including/excluding patients into our VBR analysis, we have opted to include all patients with an available Ct-value measurement within 14 days prior to the index date within our VBR analysis (page 7 line 150-152, page 12 line 266-269). In other words, our updated analysis no longer mandated the availability of Ct value measurements post treatment as an eligibility criterion. We have chosen this over-inclusive approach over the suggested approach to censor patients who do not have a Ct value measurement post treatment as our pre-index date data does not allow us to predict the probability that any given patient would eventually be censored for not having a post-treatment Ct value measurement within 28 days of the index date.

To avoid reintroducing any post-index date data for censoring purposes, we thus opted to only rely on pre-index date data (i.e. availability of a Ct-value 14 days pre-index date) to filter subjects into our study. Although our approach would necessarily lead to a dilution of the risk of VBR in both the early and late treatment groups (due to the presence of many patients who did not receive a RT-qPCR test post treatment), we respectfully submit that the fact that we continued to detect a statistically significant risk of VBR even after including so many “superfluous” subjects with a dilutive effect on our effect estimates reinforces the validity of our estimates of the risk of VBR from early nirmatrelvir/ritonavir initiation.

4. While the authors demonstrate successful balance at baseline, it remains unclear whether the IPCW sufficiently emulated the censoring at random for the two emulated trials. To substantiate the validity of the VBR analyses within the emulation framework, it's important for the authors to establish the ability to emulate not only the randomization of early versus late treatment initiation but also the randomization of the presence or absence of a Ct value.

Response 2.4: Thank you again for your comments. As discussed above, the presence or absence of a Ct value post treatment is no longer used as an inclusion criterion for our VBR analysis. Our current inclusion criteria – the presence of a Ct value within 14 days pre-index date – is known for certain at the point of treatment assignment (i.e. time zero) and thus does not require randomisation.

REVIEWERS' COMMENTS

Reviewer #2 (Remarks to the Author):

My comments have been fully addressed given the limitation of the data. I thank the authors for their efforts on making the study much more rigors compared to the initial submission. I have no further comment.

Reviewer #2 (Remarks to the Author):

My comments have been fully addressed given the limitation of the data. I thank the authors for their efforts on making the study much more rigors compared to the initial submission. I have no further comment.

Response 2.1: Thank you for your helpful comments and support.